# Birinapant Reshapes the Tumor Immunopeptidome and Enhances Antigen Presentation

**DOI:** 10.3390/ijms25073660

**Published:** 2024-03-25

**Authors:** Weiyan Zhang, Shenghuan Sun, Wenyuan Zhu, Delan Meng, Weiyi Hu, Siqi Yang, Mingjie Gao, Pengju Yao, Yuhao Wang, Qingsong Wang, Jianguo Ji

**Affiliations:** 1State Key Laboratory of Protein and Plant Gene Research, School of Life Sciences, Peking University, Beijing 100871, China; 1901110510@pku.edu.cn (W.Z.);; 2Bakar Computational Health Sciences Institute, University of California, San Francisco, CA 94143, USA; shenghuan.sun@ucsf.edu

**Keywords:** immunopeptidomics, Birinapant, tumor, antigen presentation, neoantigen, mass spectrometry, MHC

## Abstract

Birinapant, an antagonist of the inhibitor of apoptosis proteins, upregulates MHCs in tumor cells and displays a better tumoricidal effect when used in combination with immune checkpoint inhibitors, indicating that Birinapant may affect the antigen presentation pathway; however, the mechanism remains elusive. Based on high-resolution mass spectrometry and in vitro and in vivo models, we adopted integrated genomics, proteomics, and immunopeptidomics strategies to study the mechanism underlying the regulation of tumor immunity by Birinapant from the perspective of antigen presentation. Firstly, in HT29 and MCF7 cells, Birinapant increased the number and abundance of immunopeptides and source proteins. Secondly, a greater number of cancer/testis antigen peptides with increased abundance and more neoantigens were identified following Birinapant treatment. Moreover, we demonstrate the existence and immunogenicity of a neoantigen derived from insertion/deletion mutation. Thirdly, in HT29 cell-derived xenograft models, Birinapant administration also reshaped the immunopeptidome, and the tumor exhibited better immunogenicity. These data suggest that Birinapant can reshape the tumor immunopeptidome with respect to quality and quantity, which improves the presentation of CTA peptides and neoantigens, thus enhancing the immunogenicity of tumor cells. Such changes may be vital to the effectiveness of combination therapy, which can be further transferred to the clinic or aid in the development of new immunotherapeutic strategies to improve the anti-tumor immune response.

## 1. Introduction

Immunotherapy has become a hotspot in tumor treatment owing to its ability to harness the immune system to kill cancer cells [1,2,3]. For example, immune checkpoint blockade (ICB) therapy, which has been developed in recent years, uses inhibitors (ICIs) to block the interaction between checkpoints—including Programmed Cell Death 1 (PD-1) and Cytotoxic T Lymphocyte-Associated protein 4 (CTLA-4)—and their ligands to reactivate T cells [4,5]. Essentially, ICB therapy still relies on the patient’s own immune system—in which major histocompatibility complex (MHC) proteins present peptides to T cells and activate a specific immune response—to kill tumor cells [6,7]; therefore, it is minimally toxic to normal cells and greatly improves prognosis, as evidenced by clinical trials [8,9]. However, ICIs have limited efficacy in patients whose immune systems are impaired and unable to recognize and kill tumor cells. Studies have shown that ICI resistance can be caused by low tumor mutation burden (TMB), immunoediting (the loss of highly immunogenic antigens or mutations), or reduced expression of genes involved in the antigen presentation pathway [10,11,12,13], which limits their clinical application.

One feasible way to overcome ICI resistance is to upregulate the expression of MHC, inducing increased processing and presentation of a wider variety of antigens to elicit specific T-cell immune responses [14,15]. MHC class I proteins are distributed on the surface of all mammalian nucleated cells, which are also referred to as human leukocyte antigen class I (HLA-I) proteins in humans. Mistranslated or inactivated proteins are degraded into peptides within the proteasome and transferred to the endoplasmic reticulum (ER) by the transporter associated with antigen processing (TAP). Processed peptides can be loaded onto nascent HLA-I and transported to the cell surface after passing through the Golgi [16,17]; the peptide-loaded MHC (pMHC) can be recognized by CD8+ T cells specifically. In particular, there are a number of immunogenic peptides in tumors, often derived from cancer/testis antigens (CTAs) or tumor-specific antigens (TSAs) containing mutations (referred to as neoantigens), that can present their abnormal targets to T lymphocytes and activate an anti-tumor immune response [18,19,20,21]. Thus, these presented antigens (also called immunopeptides) determine the immunogenicity of cells and play an important role in immune recognition and surveillance.

Neoantigens display strong immunogenicity and have been successfully used to create vaccines that slow the development of many types of tumors, indicating their potential for use in clinical treatment [22,23,24]. One systematic process based on sequencing and prediction algorithms has been developed for the discovery of tumor mutations, accompanied by the prediction and validation of neoantigens [25,26]. However, this approach generates a vast number of candidates, but the truly immunogenic parts only constitute a small proportion; therefore, the screening of functional neoantigens relies heavily upon complex downstream immunogenicity validation [27,28]. Another direct neoantigen identification method based on mass spectrometry (MS), referred to as immunopeptidomics, displays advantages in addressing this problem since it significantly reduces the number of neoantigens that need to be experimentally verified [29,30,31,32,33]. By acquiring pMHC and then separating the mixture, the components of immunopeptides can be obtained and subjected to MS, thus revealing their sequence and potential neoantigens. This strategy has significantly improved the efficiency of neoantigen identification [29,34,35].

Birinapant is a mimetic of the second mitochondrial activator of caspases (SMACs). Previous studies on Birinapant focused on the activation of apoptotic pathways. In brief, Birinapant specifically binds to the baculoviral IAP repeat domain of inhibitor of apoptosis proteins (IAPs) through the bivalent AVPI tetrapeptide motif, then activates caspases and promotes the formation of apoptosome, and at the end induces cell apoptosis. On the other hand, Birinapant also promotes the recruitment of NIK and activates the non-canonical nuclear factor-kappa B (Nf-κB) pathway. Birinapant has been shown to inhibit tumor cell proliferation by inducing apoptosis in a variety of tumor cell lines and xenograft models. For example, D L Zhu et al. found that Birinapant inhibits the invasion and proliferation of gastric cancer cells MGC-803 by promoting apoptosis [36]. Jun Ding et al. found that Birinapant can promote apoptosis and inhibit the invasion of liver cancer cells Huh7 and HepG2 [37]. Najoua Lalaoui et al. found that Birinapant inhibits the proliferation of TNBC cells and PDX models through caspase-dependent apoptosis [38]. Birinapant also functions in some combination drug strategies to enhance efficacy. Xuemei Xie et al. found that Birinapant targets IAP in a variety of TNBC cells and xenograft models, inducing cell apoptosis, thereby enhancing the anti-tumor efficacy of gemcitabine in TNBC [39]. David Cerna et al. found that Birinapant can also enhance the radiosensitivity of glioblastoma [40]. In summary, for Birinapant, specific recognition and inactivation of IAP is a common feature; this is also a general understanding for many researchers on the drug function of SMAC mimetics.

However, a recent study showed that Birinapant could up-regulate MHC individually in numerous cell lines and enhance the efficacy of anti-tumor immunotherapy when combined with ICI [41]. Considering that MHC plays a key role in antigen presentation, it is reasonable to suggest that Birinapant may improve the antigen presentation of tumor cells, rendering them more easily recognized by T lymphocytes; nevertheless, the specific mechanism remains elusive.

Accordingly, we used immunopeptidomics in combination with genomics and proteomics strategies to explore the mechanism underlying the regulation of tumor immunotherapy by Birinapant from the perspective of antigen presentation. Treatment of HT29 and MCF7 cells with Birinapant resulted in increased diversity and abundance of the immunopeptidome, including CTA peptides and neoantigens. Moreover, it was verified that neoantigens possess the ability to activate T lymphocytes. Furthermore, how Birinapant influences immunopeptidomes in vivo was demonstrated using HT29 cell-derived xenograft (CDX) models. In conclusion, our data offer new understanding of the effect of Birinapant on the anti-tumor immune response, in addition to providing a reference for anti-tumor medication and aiding in the development of clinical treatments for cancer.

## 2. Results

Combining multiple omics methods, including genomics, proteomics, and immunopeptidomics, we designed the following pipeline to evaluate the influence of Birinapant on antigen presentation (Figure 1). Specifically, we chose two different tumor cell lines: HT29 colorectal cancer cells and MCF7 breast cancer cells. Firstly, cells were divided into two groups, one treated with DMSO (CTRL) and one treated with Birinapant (BIR). After treatment, the immunopeptidome and proteome were identified. Whole-exome sequencing (WES) was performed to obtain mutational information and the HLA-I type, and MS data were acquired by DIA. Secondly, we investigated antigens with potential value from our data, for example, CTA peptides and neoantigens, and also evaluated the immunogenicity of these neoantigens by enzyme-linked immunospot (ELISpot) and flow cytometry (FCM). Thirdly, we constructed HT29 cell-derived xenograft (CDX) models to further explore the variation in antigen presentation changes with Birinapant treatment time in vivo.

### 2.1. Birinapant Reshapes the Immunopeptidome In Vitro

Previous studies have focused on the mechanism of Birinapant as an inhibitor of IAP [42,43], neglecting its effects on antigen presentation. Recently, Gu et al. found that Birinapant could up-regulate HLA [41]. To verify this discovery, we evaluated the expression of HLA-I following Birinapant treatment by FCM, which revealed that Birinapant induced significant upregulation of HLA-I in a number of tumor cell lines (Appendix A), indicating its potential effect on antigen presentation.

We chose two frequently used cell lines, HT29 and MCF7, and compared the differences in the immunopeptidome between biological triplicate CTRL and BIR samples. Overall, a greater number of immunopeptides were identified in BIR, with an average 1.23-fold increase in HT29 and a 1.36-fold increase in MCF7 (Figure 2A,B and Appendix A). Considering that peptides may be derived from contaminants or interference factors during the experiments, we screened the peptides identified at least twice in the triplicate samples (Appendix A) for further analysis. To assess the quality of our data, we analyzed the length distribution and predicted HLA-I binding. All data show similar characteristics of length distribution (Figure 2C), which is in accordance with published articles [30,31]. WES identified five and four HLA-I alleles in HT29 and MCF7 cells, respectively, and HLA-I binding was predicted by NetMHCpan4.1. Most HLA-I alleles did not exhibit any difference following Birinapant treatment (Figure 2D,E and Appendix A); only HLA-B*35:01 and HLA-B*44:03 in HT29 cells showed a higher binding percentage (Figure 2D).

The abundance of immunopeptides reflects the expression level of corresponding pMHC on the cell surface to some extent, and the coverage rate of source proteins reflects the diversity of antigen presentation [44,45]. To evaluate the changes after Birinapant treatment, we used label-free quantitation (LFQ) to collect the relative abundance of immunopeptides and source proteins. In HT29 cells, we found an average of 3669 and 4594 immunopeptides in CTRL and BIR, respectively. In MCF7 cells, the corresponding quantity was 2155 and 3057 (Figure 3A). At the same time, we asked the question of whether these extra peptides were derived from extra proteins to demonstrate that a greater number of proteins took part in antigen presentation. In fact, there was a 1.43-fold increase in quantitated proteins in HT29 cells (from 1549 to 2215) and a 1.30-fold increase in MCF7 cells (from 1370 to 1782) in BIR compared with CTRL (Figure 3B), suggesting an improvement in antigen presentation diversity. Next, we compared the abundance of immunopeptides and proteins. Initially, we hardly observed any difference in the abundance of whole immunopeptides or proteins between BIR and CTRL in either HT29 (Appendix A) or MCF7 (Appendix A) cells. We suppose that the immunopeptides appearing only in BIR may exhibit a lower abundance in comparison with shared immunopeptides, leading to a lower overall abundance in BIR. To verify our hypothesis, we compared only the peptides that were shared between CTRL and BIR (1942 peptides in HT29 and 1325 peptides in MCF7). The results show an apparent increase in immunopeptides and proteins in BIR in both HT29 and MCF7 cells (Figure 3C–F and Appendix A), indicating an improvement in the presentation of preexisting immunopeptides after Birinapant treatment. In addition, we also compared the peptide quantity for every source protein to reflect the degree of coverage. In both HT29 and MCF7 cells, the number of proteins that contained only one peptide decreased in BIR, which accounted for the largest proportion of our data. Other groups showed a small rise (Figure 3G,H), indicating an increase in the degree of protein coverage. Essentially, Birinapant reshaped the immunopeptidome with respect to both diversity and abundance.

To investigate which biological pathways were influenced by Birinapant and reflected in antigen presentation, Metascape (https://metascape.org, accessed on 31 October 2022) [46] was used for Gene Ontology (GO) analysis. On average, 78.50% of immunopeptides in CTRL reappeared in the corresponding BIR (Appendix A), while 39.96% of immunopeptides in BIR were specific, further indicating elevated diversity in the immunopeptidome after Birinapant treatment. On average, 89.7% of proteins were shared between CTRL and the corresponding BIR (Appendix A), and 36.01% of proteins were specific to BIR. For the GO analysis specific to CTRL or BIR, in HT29 cells, “Metabolism of RNA”, “Cell Cycle”, and “Cellular protein catabolic processes” occupied the top positions in BIR (Appendix A). Similarly, in BIR of MCF7 cells, “Metabolism of RNA” and “Protein catabolic processes,” also had a high *p*-value (Appendix A), suggesting that these pathways may be activated by Birinapant and related proteins subsequently presented by antigen presentation machinery (APM) after performing their function. On the other hand, we failed to find consistent pathways in the CTRL of HT29 and MCF7 cells (Appendix A). Considering that Birinapant induces apoptosis and activation of the Nf-κB pathway [43], we suspected that these intracellular changes are accompanied by the activation of RNA metabolism and protein catabolic processes, the subsequent degradation of related proteins, and the presentation of their peptides.

Next, we sought to account for changes in the immunopeptidome through the canonical proteome using sixplex tandem mass tag (TMT)-based quantitative proteomics. For each cell line, a set of sixplex tags marked three CTRL and three BIR samples. APM-related proteins were upregulated after Birinapant treatment. In both cell lines, HLA-I was upregulated by an average of 1.30 times, TAP was upregulated by 1.15 times, and proteasome subunit beta (PSMB) was upregulated by 1.16 times (Appendix A). This feature can also be demonstrated by QPCR (Appendix A). Based on the abundance variation in certain proteins following Birinapant treatment, as exhibited in the proteome and immunopeptidome, a connection could be built between the two omics datasets. We screened proteins that had quantitative information in both datasets and calculated the fold change. In total, we obtained 868 and 434 proteins with complete abundance data in HT29 and MCF7 cells, respectively. However, the changes in the proteome could not explain the significant changes in the immunopeptidome since there was no significant correlation between the proteome and immunopeptidome (Appendix A, *p* > 0.05). This is not surprising because there was no direct correlation between the abundance of proteins and whether they could be presented by the APM. Furthermore, the HLA-I binding motif determined by the HLA-I allele also decides which peptides can be presented [29].

### 2.2. Birinapant Increases the Presentation of CTA Peptides and Neoantigens

It has been shown that CTA peptides and neoantigens have the potential to become clinical targets since they show a stronger ability to trigger the CD8+ T-cell response in comparison with other peptides [47,48]. We attempted to detect these antigens in the immunopeptidome. In HT29 cells, ten CTA peptides were identified in BIR, while four of these were identified in CTRL (Figure 4A). We compared the abundance of the four shared and six unique peptides in BIR, and the shared peptides had a higher abundance (Figure 4B). Subsequently, we compared the abundance of CTA peptides using the LFQ method. The abundance of the four shared CTA peptides was significantly higher in BIR, indicating increased presentation of these peptides induced by Birinapant (Figure 4E). We thought that this may demonstrate a general influence of Birinapant on the promotion of CTA presentation since these shared CTA peptides have a stronger ability by nature to be presented. Considering that immunogenicity is an index for evaluating the ability of antigens to activate T lymphocytes, we acquired the immunogenicity scores of CTA peptides using the MHC-I immunogenicity module of T Cell Epitope Prediction provided by the Immune Epitope Database (IEDB) Analysis Resource. Six peptides showed strong immunogenicity (score > 0.1), four of which were found only in BIR (Table 1), indicating that CTA peptides presented after Birinapant treatment had better immunogenicity.

We performed the same analysis for CTA peptides in MCF7. Specifically, 20 CTA peptides were identified in BIR, which included all 14 CTA peptides appearing in CTRL (Figure 4C). At the same time, shared peptides had a higher abundance in comparison with specific peptides in BIR (Figure 4D). Regarding abundance, nine of the fourteen shared peptides showed a significant difference, while seven of these (77.7%) appeared at a higher abundance in BIR (Figure 4F). In terms of epitope immunogenicity prediction, seven CTA peptides showed strong immunogenicity (score > 0.1), and the top three were only found in BIR (Table 2), confirming our conclusion. Interestingly, we found eight peptides in BIR (four in CTRL) that had no record of discovery in IEDB, two of which occupied the top two positions in the immunogenicity ranking list (Table 2), indicating that Birinapant may aid in the discovery of new targets.

Using the protein database containing mutational information according to WES, we successfully identified several neoantigens, five in each cell line. In HT29 cells, four neoantigens were derived from BIR and one was derived from CTRL, but no neoantigens were shared between the two groups (Table 3). In MCF7 cells, four neoantigens were derived from BIR, two were derived from CTRL, and one was shared (Table 3). Among them, one neoantigen (EKPVHLHGPPA) belonged to insertion/deletion mutation (Indel) proteins, which are rare in immunopeptidomics studies [49]. The percentage of HLA-I binding predictions of neoantigens varied from 0.12 to 44 (a smaller number means better binding ability). The source proteins of neoantigens did not belong to common driver mutations in cancer and had no association with each other; this may be because, under the influence of immunoediting, the tumor evolves into a less antigenic tumor in order to evade immune system surveillance [50]. There was no relevant record of these neoantigens in the IEDB, indicating their individual specificity. In conclusion, Birinapant treatment increased the diversity of the presented neoantigens.

### 2.3. Neoantigens in BIR Display Stronger Immunogenicity

The quality rather than quantity of neoantigens may decide the prognosis in neoantigen-based tumor immune therapy [51], and immunogenicity is an important parameter for the quality of neoantigens. We selected the neoantigen (EKPVHLHGPPA) Neo1 from “binding percentage” and MS2 of the neoantigen. Neo1 comes from an Indel, which is relatively rare in the study of immunopeptidomes. The source protein of Neo1 is MYO1E, which serves in intracellular movements as an unconventional myosin. According to the WES sequencing results, we found that a “C” was inserted before the “A” at position 2877 of MYO1E exon, thus causing a G to R mutation after the original AAPPPP in the protein sequence, and the subsequent sequence also completely changed (Figure 5A,B). Next, to further demonstrate the existence of Neo1 in the transcriptome, we extracted the mRNA of HT29 and performed sequencing. The result showed that Indel existed in the transcript, and the presence of double peaks indicated that this mutation was non-homozygous (Figure 5C). 

We searched this mutation in the Uniprot protein database and failed to find consistent information, and in IEDB, there was also no record of this peptide. In order to ensure the correct identification of this neoantigen, we synthesized this peptide in vitro and detected it on the same mass spectrometer. The MS2 results of the synthesized peptide and presented peptide were relatively similar, indicating that our identification was credible (Figure 5D–F). Next, we used Elispot to verify the immunogenicity of Neo1. Based on the HLA-I alleles of cell lines, PBMCs from healthy donors that had at least three matched alleles were used (Appendix A). According to a published study [52], we performed three rounds of pre-stimulation (three days per round); however, after coincubation, Neo1 failed to display the ability to activate T lymphocytes (Figure 5G). Considering that healthy donors have not been previously exposed to the candidate and need to generate antigen-specific T lymphocytes de novo, we appropriately lengthened the coincubation time of peptides and PBMCs to avoid false negative results caused by inadequate stimulation. After five rounds of stimulation (three days per round), Neo1 showed a good ability to induce IFN-γ secretion by T lymphocytes (Figure 5H,I). We demonstrate that Neo1 induced by Birinapant was immunogenic and sufficient to elicit T cell immune responses; meanwhile, compared with memory recognition, de novo alloantigens’ recognition of T lymphocytes may require more time for stimulation, as five instead of three rounds of coincubation successfully induce T lymphocyte activation.

### 2.4. Birinapant Promotes Antigen Presentation In Vivo

To explore the effect of Birinapant on antigen presentation in vivo, we constructed CDX models in immunodeficient mice. To obtain sufficient tumor samples for immunopeptidomes, we chose HT29 for model construction (MCF7 is more difficult to grow subcutaneously). Mice were divided into four groups, three of which were administered Birinapant for 3 days (BIR-3), 6 days (BIR-6), or 9 days (BIR-9), and the fourth was administered saline for 6 days as a control (CTRL). Each treatment cycle lasted three days, and after three cycles, mice were sacrificed and tumors were removed for immunopeptidomics analysis. Overall, the growth of the mice was stable, with no significant weight loss during the drug infusion period, while there was a clear reduction in the size of the tumor after Birinapant treatment (Figure 6A,B). Since Birinapant had the function of promoting cell apoptosis, it was expected. 

We randomly selected three samples from each group for immunopeptidomics analysis. MS results were analyzed using the PEAKS Studio Xpro software. On average, 2711, 2908, 4531, and 4844 peptides were identified in CTRL, BIR-3, BIR-6, and BIR-9, respectively (Figure 6C). For the source proteins, the corresponding data were 1544, 1919, 2564, and 2938, respectively (Figure 6D). Similarly, we screened the peptides that were identified at least twice in the biological triplicate datasets for further analysis (Figure 6E–H), with the number of screened immunopeptides being 2375, 3091, 4306, and 4432, respectively (Figure 6I). With prolonged treatment of Birinapant, the number of immunopeptides and source proteins continued to increase in vivo. We then compared the length distribution and HLA-I binding prediction of the immunopeptidome in the four groups. For length distribution, data in all groups met the basic features of the immunopeptidome, and there was little difference between groups (Figure 6J). Regarding HLA-I binding, although the trend of change in each group was not entirely consistent, for example, in BIR-6, the percentage of peptides predicted to bind HLA-A2403 was lower, and the differences between groups were not significant (Figure 6K and Appendix A).

Next, we compared the abundance of immunopeptides. A total of 1302 peptides were shared among all four groups (Figure 7A) and showed a lower abundance in CTRL, with the average abundance following logarithm transformation reaching 5.719, while in the other three groups treated with Birinapant, the abundance was 6.375, 6.225, and 6.579, respectively (Figure 7B). These findings indicate that Birinapant improved the abundance of presented antigens in vivo; however, there was no positive correlation with treatment time (abundance in BIR-6 was lower than abundance in BIR-3). We also compared the variation in abundance of source proteins to ensure that Birinapant indeed induced more frequent protein processing. Consistently, the average abundance of the 1072 shared source proteins in CTRL, BIR-3, BIR-6, and BIR-9 was 5.928, 6.560, 6.440, and 6.781, respectively (Figure 7C and Appendix A). Meanwhile, the coverage rate of proteins also indicates that both the diversity and abundance of the immunopeptidome improved following treatment with Birinapant in vivo (Figure 7D). In conclusion, we found that Birinapant could also reshape the immunopeptidome in diversity and abundance in vivo, although this influence was not strictly positively correlated with treatment time.

Subsequently, we performed GO analysis of source proteins to gain a systematic understanding of the presentation tendency. Firstly, we focused on proteins that appeared in all three groups treated with Birinapant but not in CTRL. These 500 proteins may have a lower tendency to be presented under normal conditions, but their presentation is promoted by Birinapant (Appendix A). Using Metascape to perform functional clustering, we found that these clusters still included “Metabolism of RNA”, “Cell Cycle”, and other related pathways (Appendix A) that were seen in HT29 cells following Birinapant treatment. Next, we examined the proteins that appeared in any BIR group (BIR-3, BIR-6, or BIR-9), revealing 2048 proteins, which showed similar clusters to those of the 500 shared proteins (Appendix A). Furthermore, in comparison with CTRL, specific proteins in each BIR group (925 proteins in BIR-3, 1245 proteins in BIR-6, and 1558 proteins in BIR-9, Appendix A) also showed many consistent clusters (Appendix A). Taken together, these data further demonstrate that Birinapant deeply affects antigen presentation in these pathways.

Next, we considered whether certain immunopeptides could be consistently enhanced by Birinapant for presentation. Based on the 1302 shared peptides in the four groups, the variation in abundance of each immunopeptide with prolonged Birinapant treatment time was evaluated (BIR-3/CTRL, BIR-6/BIR-3, and BIR-9/BIR-6). Interestingly, no peptides were at the top 200 positions in all three comparisons (Figure 8A), indicating that Birinapant may influence antigen presentation in specific pathways rather than specific proteins or peptides. We then focused on the top 200 peptides in the average ranking list for functional clustering, and “Metabolism of RNA” still occupied the top position (Appendix A), further supporting our conclusion. However, we discovered that although these peptides showed a similar total binding percentage to the immunopeptides identified in the four groups, a degree of difference existed in specific HLA-I alleles. For example, HLA-A0101 and HLA-B4403 showed a higher percentage of these shared peptides, while HLA-A2403 showed a lower percentage (Figure 8B). Further studies were required to explain this change.

Regarding CTA peptides, we identified 3, 9, 14, and 11 in CTRL, BIR-3, BIR-6, and BIR-9, respectively (Table 4). Only BIR-6 contained all three CTA peptides that appeared in CTRL, and this group also contained the largest quantity of CTAs. Meanwhile, in comparison with CTRL, CTA peptides exhibited a higher abundance in the three BIR groups (Figure 8C). These results indicate that Birinapant improves the quality and quantity of CTA peptides in vivo, promoting their presentation and recognition. Moreover, we searched for neoantigens. In BIR-3, we found “LRVQLHLKK”, a peptide derived from DNAH3 with a P1197L mutation (in the fifth position of the peptide). In BIR-6 and BIR-9, we found another neoantigen, “SEIRHTANRW” (SEI), which was derived from HSPB1 with a D93N mutation (in the eight position of the peptide). Although we only found two neoantigens in the CDX models, much less than those found in cells, SEI had the highest −10lgP value, indicating the highest credibility (Table 5). Additionally, SEI was predicted to have strong immunogenicity (score = 0.22619) and was also identified in two experiments (BIR-6 or BIR-9); by contrast, the other neoantigens only appeared in one experiment, further indicating the application value of SEI. These data demonstrate that Birinapant promotes the diversity and abundance of presented CTA peptides and neoantigens, while these peptides are predicted to be more immunogenic.

Lastly, we compared the immunopeptidome between the HT29 CDX models and cells and found that there was not a high degree of overlap. For example, only 20.20% of peptides in CTRL-CDX were shared with CTRL-cell, and in the BIR groups, this similarity increased to 37.75%, 37.06%, and 39.19% for BIR-3, BIR-6, and BIR-9, respectively (Appendix A). It is reasonable to suggest that this may be due to the different culture environments in vitro and in vivo. Interestingly, for CTA peptides identified in both cells and CDX models, seven of the ten CTAs reoccurred in BIR-6 and eight reoccurred in BIR-9 (Table 2 and Table 4). Although the overlap between the immunopeptidome of cells and CDX was low, the presentation of CTAs appeared to be conserved.

## 3. Discussion

Here, we reveal that Birinapant could enhance tumor antigen presentation from the perspective of the immunopeptidome. Specifically, both in vitro and in vivo, Birinapant reshaped the immunopeptidome in diversity and abundance, induced the presentation of antigens derived from specific pathways such as “Metabolism of RNA”, and promoted the presentation of immunogenic CTA peptides and neoantigens, thereby optimizing the immunogenicity of tumor cells.

Neoantigens contain tumor-specific mutations and have strong immunogenicity. Accordingly, dendritic cell vaccines that present neoantigens, in addition to RNA or peptide vaccines, have been developed to activate T lymphocytes and achieve specific tumoricidal effects. However, patients with low TMB or low HLA expression in tumors caused by immunoediting possess extremely few tumor-specific T lymphocytes; therefore, it is difficult for these patients to identify neoantigens, and they are typically resistant to these vaccines. In such patients, attempting to activate HLA expression levels to promote the antigen presentation of tumor cells may prove to be a valid approach. For example, studies have shown that interferon (IFN)-γ can upregulate tumor HLA expression, promote the expression of genes related to antigen presentation pathways, and enhance antigen presentation [53,54,55]. Further reports have demonstrated that IFN-γ can inhibit tumor growth in vivo [56,57]. Nevertheless, IFN-γ can cause cytokine storms in vivo and result in serious side effects, limiting its further development as an antitumor drug. In the present study, we found that Birinapant could upregulate tumor HLA-I and enhance antigen presentation in mice without causing significant weight loss.

Birinapant, as an SMAC mimetic, has attracted increasing attention due to its ability to inhibit the expression of IAP and suppress tumor growth, and many related clinical trials have been carried out. Owing to the rapid development of tumor immunotherapy, the regulation of HLA proteins and antigen presentation by Birinapant has also been focused upon. Similarly, other members of the SMAC mimetic group, for example, ASTX660 and APG 1387, have been reported to upregulate HLA-I and HLA-DR, respectively [58,59]. These data indicate that many SMAC mimetics may possess the ability to affect antigen presentation, and our study provides a reference for relevant research.

The immunopeptidome consists of all the immunopeptides presented by HLA proteins. Although most immunopeptides are derived from normal proteins originally existing in cells and unable to elicit an immune response, a small proportion of immunopeptides are derived from CTAs or TSAs; therefore, they are more easily recognized as non-self-antigens by T lymphocytes or antigen-presenting cells, thereby enhancing anti-tumor immunity. In the past, many excellent studies have taken advantage of immunopeptidomics to directly identify antigens as personalized therapeutic targets for clinical treatment [30,31,60,61]. However, our work pioneers the use of immunopeptidomics to explore the regulatory mechanism of Birinapant in anti-tumor immunity. We believe that Birinapant promotes the immune presentation of CTA peptides and neoantigens, which improves the immunogenicity of tumors and may render them more sensitive to immunotherapy. Conventional genomics, transcriptomics, and proteomics are excellent for elucidating drug targets; nevertheless, these strategies are not ideal for identifying peptide targets that play a key role in immune responses. Immunopeptidomics has a unique advantage in this field, providing a large amount of peptide information to reflect the presentation preference of tumors and aid further understanding of the tumor suppressor function of drugs.

Neoantigens are potential targets for tumor therapy. A very important step in the discovery of neoantigens is the verification of their immunogenicity, which determines the ability of neoantigens to elicit an immune response [20,26]. Here, we used ELISpot to reliably verify the immunogenicity of neoantigens. Considering that it was impossible to obtain peripheral blood corresponding to the cell lines, we selected peripheral blood derived from healthy donors for use in our experiments. However, since this would likely lead to de novo T-cell immune responses to the neoantigens, requiring more time for the proliferation of cytotoxic T lymphocytes in comparison with a memory response, we extended the incubation time of peptides and PBMCs. These data show that five rounds of stimulation produced a higher level of T-cell activation than three rounds of stimulation, confirming our assumption. Due to the limited amounts of PBMCs, we could not perform immunogenicity verification of all CTAs and neoantigens.

We compared the immunopeptidomes of HT29 in vivo and in vitro. We found that the overlap was small. We believed this may be mainly caused by differences in the culture environment. One possible reason is that tumor cells face different pressures in vivo and in vitro, which may affect the immunopeptidomes of cells. It should be pointed out that all our immunopeptidome results come from three biological replicates, and the repeatability within the group seemed good, so the low overlap is unlikely to be caused by erroneous identification. We found in one study that the immunopeptidomes of cells and xenograft models were highly reproducible [62]. Therefore, it is also possible that the characteristics of the cells cause this phenomenon. In addition, it may be related to the experimental method of immunopeptidomics. We found that Heather JM et al. used 1 × 10^9^ cells and 1 g tumor tissue for immunoprecipitation, which greatly exceeded the amount of ours, but the number of immunopeptides they identified seemed smaller, especially in MDA-MB-436 and Colo205 cells. This may mean that a large number of peptides with low abundance were lost. Overall, this question still needs to be explored in more cell models to draw more credible conclusions.

To take a long view, we believe that Birinapant has the potential to be used as an adjuvant, enhance the immunogenicity of tumor cells, and make patients benefit more from immunotherapy. Currently, as the most effective immunotherapy, ICB still has limited efficacy in some tumors (such as glioblastoma and pancreatic cancer) [63]. Gu et al. have demonstrated that Birinapant combined with ICB can further inhibit tumor growth [41], which suggests a possible strategy to address the poor performance of ICB in some cases. On the other hand, Birinapant enhances the antigen presentation pathway of tumor cells. For CAR-T or TCR-T therapy, this may lead to better recognition. Furthermore, neoantigen vaccines and dendritic cell vaccines [64] may also benefit from better antigen presentation by tumor cells.

Certain deficiencies and limitations exist in the present study, and further investigation is required. Firstly, SMAC mimetics are a large class of drugs, some of which have entered clinical trials and shown excellent anti-tumor effects; meanwhile, regarding safety, they are mainly grade-1–2 events, with sporadic grade-3 events [65,66,67,68]. Here, we only explored the mechanism underlying the effect of Birinapant on anti-tumor immunity, and it remains to be elucidated whether other similar drugs upregulate HLA-I and to what extent. Moreover, although the immunopeptidome contains thousands of peptides, the majority are not immunogenic. Neoantigens have not undergone positive or negative selection and are therefore ideal targets for immunotherapy. Nevertheless, a common problem in our work and numerous other immunopeptidomics studies is the low discovery rate of neoantigens, which results in a limited selection of candidate targets for subsequent experimental verification and clinical trials, further limiting the application of immunopeptidomics. Furthermore, although we demonstrated that, compared with Birinapant, the effect of DMSO on HLA was negligible by Western blot and QPCR (Appendix A), the effect on immunopeptidomes of DMSO is actually unclear and remains to be elucidated.

In conclusion, this work pioneers the use of immunopeptidomics in conjunction with cell lines and CDX models to explore the regulatory mechanism underlying the effect of Birinapant on tumor immunity. Birinapant increased the diversity and abundance of immunopeptides, in addition to significantly promoting the presentation of immunogenic CTA peptides and neoantigens not only in quantity but also in quality and abundance, which improved the immunogenicity of tumors. This work is expected to be further extended to the clinic with a view to developing better tumor immunotherapy and providing a reference for tumor combination therapy.

## 4. Materials and Methods

### 4.1. Instrument, Software and Reagent

Orbitrap Fusion^TM^ Lumos^TM^ Tribrid^TM^, ThermoFisher, Waltham, MA, USA

DMEM, HycloneTM GE Healthcare Life Sciences, Marlborough, MA, USA

AIM-V, ThermoFisher, Waltham, MA, USA

Fetal bovine serum, Gemini, Sacramento, CA, USA

Protein-A Sepharose, Invitrogen, Waltham, MA, USA

Birinapant, MCE, Shanghai, China

Bradford Protein Assay Kit, Tiangen, Beijing, China

cDNA Synthesis Kit, Bio-Rad, Hong Kong, China

human IFN-γ-precoated ELISpot Kit, Dakewe, Shenzhen, China

PEAKS Studio Xpro, Bioinformatics Solutions Inc., Waterloo, Ontario, Cananda

Tree Star FlowJo^®^ V10, Becton, Dickinson & Co., Franklin Lakes, NJ, USA

### 4.2. Cell Culture

HT29, MCF7, MIA PACA-2, and LOVO cell lines were purchased from the American Type Culture Collection (ATCC) and maintained in Dulbecco’s modified Eagle medium (DMEM, Hyclone^TM^ GE Healthcare Life Sciences, Marlborough, MA, USA) supplemented with 10% (*v*/*v*) fetal bovine serum (FBS, Gemini #900-108) at 37 °C and 5% CO_2_. HT29 and LOVO belong to colorectal cancer, and MCF7 belongs to breast cancer, which are the two most common types of cancer. MIA PACA-2 belongs to pancreatic cancer, which is the cancer type with the worst prognosis. All cells were confirmed to be free of mycoplasma contamination. Birinapant (MCE, shanghai, China, #HY-16591) was dissolved in 50% DMSO and stored at −80 °C. Cells were grown to 80–90% confluence; Birinapant was added to the medium for 48 h; and then cells were lysed for subsequent experiments. Cell lysates were stored at −80 °C.

Peripheral blood mononuclear cells (PBMCs) derived from healthy donors were purchased from Milestone Biotechnologies. PBMCs were maintained in AIM-V (ThermoFisher Waltham, MA, USA #0870112DK) medium supplemented with 10% (*v*/*v*) FBS at 37 °C and 5% CO_2_.

### 4.3. Whole-Exome Sequencing and HLA-I Typing

5 × 10^7^ cells were used to perform whole-exome sequencing (WES) by Novogene and generate a cell line-specific database. The Hiseq-PE150 sequencing platform was used for double-ended sequencing in a 150-bp format, achieving a minimum coverage of 10× for more than 99% of the genome. The files were then converted using the BCL2FastQ v1.8.4 software (Illumina). After the removal of the low-quality reads, the Burrows–Wheeler Aligner (BWA) was compared with the Human Reference Gene Bank (HS37D5), and the sambamba tool was used to remove the repeat reads. HLA-I typing analysis was performed after WES. For HT29, the HLA-I alleles were as follows: HLA-A*01:01; HLA-A*24:03; HLA-B*35:01; HLA-B*44:03; and HLA-C*04:01. For MCF7, the HLA-I alleles were: HLA-A*02:01; HLA-B*18:01; HLA-B*44:02; and HLA-C*05:01.

### 4.4. HLA-I Immunoprecipitation

This experimental protocol was generally based on the procedure formulated by Bassani–Sternberg [30]; however, it has been slightly modified according to the procedures described by Anthony W. Purcell [69]. 

For the W6/32 antibody, we used W6/32 hybridoma purchased from the China Center for Type Culture Collection (CCTCC), and cultured in 1640 medium containing 10% FBS at 37 °C and 5% CO_2_. When the cells reached 3 × 10^7^, centrifuged at 1000× *g* for 2 min and resuspended in 15 mL of CD Hybridoma Medium (Gibco #11279-023) containing 1% 200 mM GlutaMAX^TM^-1 (100X, Gibco #35050-061). Cells were added to the cell compartment of CELLine 1000 (WHEATON #WCL1000-3) to culture and generate antibodies and 1000 mL medium of were added to the medium compartment. After one week, the mediums in the cell compartment were taken out and used for antibody purification. Disposable 5 mL Polypropylene Columns (Thermo Fisher #29922) were used for purification. The medium was centrifuged at 4 °C at 8000× *g* for 5 min to remove cells, and then 1 mL of Protein-A Sepharose (Invitrogen (Waltham, MA, USA) #101042) was added before being incubated at 4 °C for 2 h. Then, it was centrifuged at 4 °C at 1500× *g* for 5 min, and the supernatant was carefully removed. Sepharose was washed by 5 mL of 100 mM Tris and 20 mM Tris in turn, then the antibody was eluted by 5 mL of Elution buffer (0.1 M acetic acid, pH 3.0). The eluate was added to 1.5 mL of 1 M Tris to keep the antibody stable. Antibody can be used for following cross-link or stored at −80 °C by adding 50% glycerin. The concentrations of antibodies were measured using Bradford Protein Assay Kit (Tiangen (Beijing, China) #PA102). A total of 400 μL of Protein-A Sepharose and 2 mg antibodies were added in columns and incubated at room temperature (RT) for 2 h, and then the liquid flowed out from the bottom. We then used 5 mL of Sodium Borate buffer (pH 9.0) to flow through the column to wash Sepharose and added 5 mL of 20 mM dimethyl pimelimidate dihydrochloride (DMP, dissolved in 0.2 M Sodium Borate buffer pH 9.0) for 30 min at RT to finish cross-linking; then, we let the liquid flow out. We added 2.5 mL 0.2 M ethanolamine and incubated it at RT for 2 h. Sepharose could be used for immunoprecipitation or stored in PBS with 0.02% NaN_3_ at 4 °C.

For immunoprecipitation, 1 × 10^8^ cells were resuspended in 5 mL lysis buffer (saw composition in 30), homogenized 3–4 times for 15 s each, ultrasonicated, and incubated at 4 °C for 1 h. Subsequently, the homogenate was centrifuged at 16,000× *g* for 40 min at 4 °C, and the supernatant was filtered through a 0.22 µm filter. The protein concentration of the lysate was then measured using the Pierce™ BCA Protein Assay Kit (Thermo Fisher Scientific, #23227). A total of 200 μL of W6/32-conjugated Protein A-Sepharose 4B^®^ (ThermoFisher #101042) was used to pull down HLA-I molecules. Reversed-phase high-performance liquid chromatography (RP-HPLC) used a reverse C18 column with an inner diameter of 4.6 mm × 50 mm to isolate immunopeptides based on a gradient from buffer A (2% ACN/0.1% TFA) to buffer B (80% ACN/0.1% TFA). The peptide fraction (before 45% buffer B) was collected for mass spectrometry.

### 4.5. Mass Spectrometry of Immunopeptidomes

Immunopeptides were loaded into a micro-flow loading pump. For the nanoLC, peptides were delivered to an Acclaim PepMap 75 μm × 2 cm NanoViper C18 and a 3 μm column (Thermo Fisher) at an 8 μL/min flow rate by EASY-nLC 1200 system. Solvent A was 0.1% formic acid, and solvent B was 0.1% formic acid in 80% acetonitrile. Mass spectrometry was performed using an Orbitrap Fusion Lumos Tribid instrument (Thermo Fisher) with the following parameters: scanning range, 300–1800 *m*/*z*; maximum sampling time, 120 ms; MS1 resolution, 120,000; highest velocity mode for the precursor ion; period, 2 s; charge state, 1–4+; high-energy collision dissociation, 28% at 2–4+ and 32% at 1+; MS2 resolution, 60,000; and a 1.2 Da separation window. The duration time was 140 min, and the gradient was 2% to 90% solvent B. 

Immunopeptides were identified using PEAKS Studio Xpro software for data-independent acquisition (DIA). Databases, including the personalized mutant gene bank based on WES, were used. Carbamidomethylation (57.021 Da), methionine oxidation (15.995 Da), and acetylation (42.011 Da) of the amino terminal were set as variable modifications. Enzyme specificity was set to no specificity. The false discovery rate (FDR) was set to 0.01. The precursor mass was set to 10 ppm, and the identification length of peptides was 8–15 aa.

For the quantification of immunopeptides, we used label-free quantification (LFQ) in PEAKS Studio X Pro software, performed after the above DIA and database matching. The LFQ method was ID-directed LFQ, and match between runs was used; the retention time shift tolerance (min) was auto-detected; the fragment mass error tolerance was set to 0.02 Da; and FDR was set to 0.01. The reference sample and training samples were auto-detected. For the peptide feature, the quality was set to ≥2 and charged between 1 and 10. For the protein, FDR was set to 0.05.

### 4.6. Immunoblotting

Proteins were separated by sodium dodecyl sulfate-polyacrylamide gel electrophoresis (SDS-PAGE) and then transferred to the polyvinylidene fluoride (PVDF) membrane in the Trans-Blot^®^ system. Membranes were blocked with skimmed milk at room temperature (RT) for 1 h and then incubated overnight with specific primary antibodies. The next day, membranes were incubated with secondary antibodies at RT for 1 h, and then immunodetection was performed using a chemiluminescence imaging system.

### 4.7. Protein Desalting

The desalting column was sequentially rinsed with 1 mL ACN and 1 mL 0.1% trifluoroacetic acid (TFA)/70% ACN, and then equilibrated with 1 mL 0.1% TFA. The sample was added to the column, and the flow through was collected; repeated once. The sample was then desalted with 1 mL of 0.1% TFA and eluted with 1 mL of 0.1% TFA/70% ACN. The eluate was concentrated for subsequent mass spectrometry.

### 4.8. Sixplex-Tandem Mass Tag (TMT)-Labeled Proteome

A total of 2 × 10^6^ cells were lysed with 60 μL 8 M urea and ultrasonic crushing, after which 3 μL 100 mM dithiothreitol (DTT) was added and incubated at RT for 30 min. Subsequently, 3 μL 200 mM iodoacetamide (IAA) was added and incubated at RT for 30 min, then another 3 μL 100 mM DTT was added and incubated at room temperature for a further 10 min. Lys-c was added at a ratio of 1:100 and allowed to digest at 37 °C for 3 h. The sample was diluted 4 times with 100 mM Tris-HCl, and then trypsin was added at a ratio of 1:50 and allowed to digest at 37 °C for 8–18 h. TFA was added at a final concentration of 0.5% for acidification, and then the sample was desalted. The concentrated sample was dissolved in 1.0 M triethylammonium bicarbonate (TEAB), and the pH was adjusted to 8.5. The TMT reagent was then added to each sample and incubated at RT for 1 h. Subsequently, 5% hydroxylamine (HDX) was added to terminate the reaction. All the samples were pooled and concentrated.

### 4.9. Horseradish Peroxidase (HRP) Fractionation and Mass Spectrometry of Proteome

The concentrated sample was dissolved in H_2_O (pH 10.0). The C18 column was rinsed with ACN, equilibrated with H_2_O (pH 10.0), and then the sample was loaded; this was repeated twice. The sample was desalted with 200 μL of H_2_O (pH 10.0) and eluted with ACN. The gradient was 10%, 12.5%, 15%, 17.5%, 20%, 22.5%, 25%, and 50%. After elution, the sample was concentrated and used directly for mass spectrometry.

Samples were loaded into a micro-flow loading pump. The nanoLC and MS parameters were the same as those mentioned above; the only difference was that the duration time was 194 min and the gradient was 6 to 90% solvent B.

Use PEAKS Studio version Xpro software to perform identification and quantification. Mass tolerance was set to 10.0 ppm, normalization was auto-finished, and FDR was set to 0.01.

### 4.10. Peptide Synthesis and Mass Spectrometry Analysis

Peptides were chemically synthesized by Science Biotechnology Co., Ltd. (Hefei China) at a purity of >98%. For neoantigens derived from single-nucleotide variants (SNVs), the corresponding wild-type peptides were synthesized simultaneously. All peptides were dissolved in 400 μL of 10% DMSO per 1 mg, combined, and desalted for mass spectrometry analysis.

### 4.11. mRNA Isolation and Reverse Transcription

Use RNA Isolation Kit V2 (Vazyme #RC112-01) to harvest the mRNA of HT29. Based on the manual, cells were lysed in Buffer RL, then used in the FastPure gDNA-Filter Column to remove gDNA. Ethanol was added to the filtrate, and FastPure RNA Column to purify mRNA. The RNA Columns were washed with Buffer RW1 and RW2, then used RNase-free ddH_2_O to elute the RNA. 

We used the cDNA Synthesis Kit (Bio-Rad (Hong Kong, China) #1708890) for reverse transcription. The mix of RNA template (1 µg) and RNase-free ddH_2_O was added to iScript Reaction Mix (5 µL) and iScript Reverse Transcriptase (1 µL), then performed the following protocol to obtain cDNA: 5 min at 25 °C, 20 min at 46 °C, 1 min at 95 °C, hold at 4 °C.

The sequencing was performed by Beijing Tsingke Biotech Co., Ltd. (Bejijng China), and the primers were:

F: GGTTATTCCAGTGGGACTCAAAA

R: TGACACTCGGTCTGAACTGGTAG

### 4.12. Detection of Neoantigen Immunogenicity by Enzyme-Linked Immuno-Spot

A total of 1 × 10^5^/well PBMCs were pre-stimulated with 10 µg/well neoantigens or wild-type peptides (corresponding non-mutation peptides for Indel, compared with negative control) and 4 IU/well IL-2 for 9 or 15 days (3 rounds or 5 rounds, 3 days per round), replenished half of medium, peptides, and IL-2 per 3 days, then PBMCs were seeded onto antibody-coated plates for subsequent processing. The human IFN-γ-precoated ELISpot Kit (Dakewe Shenzhen, China #2110005) was used according to the manufacturer’s instructions. Ice-cold H_2_O was added to lyse the cells, which were then washed and incubated with biotinylated antibody at 37 °C for 1 h. Cells were washed again, incubated with streptavidin-HRP at 37 °C for a further 1 h, washed again, and then incubated with chromogenic solution at RT in the dark for 5–30 min. Cells were washed again, allowed to dry naturally in the dark, and then the spots were counted. For the negative control, no peptide was added. For positive control, the positive stimulant phorbol myristoyl acetate (PMA) was added.

### 4.13. Flow Cytometry

Flow cytometry (FCM) was performed using the APC anti-human pan-HLA-I antibody W6/32 (Dakewe #311410). Biologically triplicated 1 × 10^5^ cells were collected, centrifuged at 500× *g* for 5 min, and resuspended in phosphate-buffered saline (PBS) containing 2% FBS. Each sample was labeled with 5 μL of W6/32 antibody for 30 min at 4 °C, washed three times with PBS, and resuspended in PBS for analysis. All experiments were performed using the BD FACSVerse™ instrument (Phoenix Instrument Platform of Peking University), and data analysis was carried out using the Tree Star FlowJo^®^ V10 software (Becton, Dickinson & Co., Franklin Lakes, NJ, USA).

### 4.14. Cell-Derived Xenograft Construction and Drug Treatment

BALB/c nude mice (6–8 weeks old) were housed in SPF animal breeding rooms. A total of 5 × 10^7^/mL HT29 cell suspension containing 20% Matrigel^®^ was subcutaneously injected to promote tumor formation. After the tumors volume reached ~270 mm^3^ (easy to observe and measure changes in volume), mice were divided into 4 groups and each group contained at least 5 mice. The control group was administered saline for 6 days (CTRL), and the treatment groups were administered Birinapant for 3 days (BIR-3), 6 days (BIR-6), or 9 days (BIR-9). Birinapant was dissolved in DMSO and mixed with 9 volumes of saline containing 12.5% Captisol^®^, which was administered to each mouse in the treatment groups by intraperitoneal injection at a dose of 30 mg/kg [37]. The CTRL group was injected with saline containing 12.5% Captisol^®^ and 10% DMSO. All mice were administered for 3 days. After the drug treatment was completed, mice were sacrificed and tumors were removed. Tumors were used directly for subsequent immunopeptidome analysis. Tumors (200 mg were used) were frozen in liquid nitrogen, then cut into small pieces, resuspended in 5 mL lysis buffer, and homogenized several times for 15 s each to form a turbid mixture. The following procedures were the same as for cells.

The animal study was reviewed and approved by the Institutional Animal Care and Use Committee (IACUC) of Peking University.

### 4.15. Statistical Analysis

All data were statistically analyzed using the GraphPad Prism 8.0.2 software. *p* < 0.05 was considered to indicate a statistically significant difference.

## Figures and Tables

**Figure 1 ijms-25-03660-f001:**
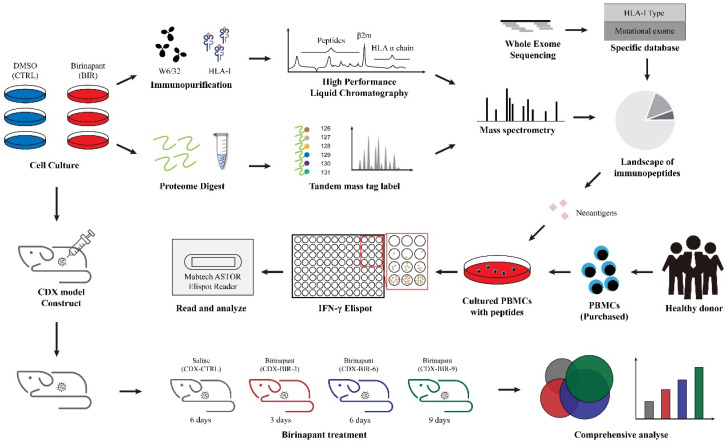
Research pipeline. MCF7 and HT29 cells were divided into two groups: one was treated with DMSO (CTRL), and was treated with Birinapant (BIR). For immunopeptidomics, W6/32 antibodies were used to immunopurify HLA-I, and then high-performance liquid chromatography (HPLC) was employed to separate peptides and HLA-I proteins. For proteomics, proteins were labeled by sixplex tandem mass tag (TMT). An Orbitrap Fusion™ Lumos™ mass spectrometer was used for mass spectrometry, and the PEAKS Studio Xpro software was used for de novo data acquisition and database search. Whole-exome sequencing (WES) was used to construct a specific database of cells containing mutations and specific HLA-I alleles. For neoantigens identified in the immunopeptidome, some were selected for immunogenicity evaluation by IFN-γ enzyme-linked immunospot (ELISpot). HT29 cells were used to construct cell-derived xenograft (CDX) models, which were divided into four groups treated with saline for 6 days (CDX-CTRL), Birinapant for 3 days (CDX-BIR-3), 6 days (CDX-BIR-6), and 9 days (CDX-BIR-9), respectively.

**Figure 2 ijms-25-03660-f002:**
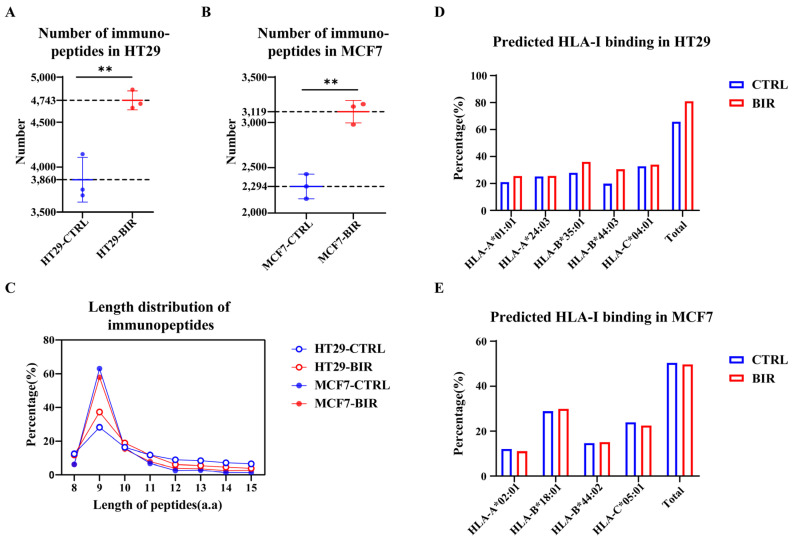
Basic characteristics of the immunopeptidome in cell lines. (**A**) The number of immunopeptides identified in HT29-BIR was increased by an average of 1.23-fold (from 3860 to 4743) in comparison with HT29-CTRL. (**B**) The number of immunopeptides identified in MCF7-BIR was increased by an average of 1.36-fold (from 2294 to 3119) in comparison with MCF7-CTRL. (**C**) Length distribution of our immunopeptides met the basic characteristics reported in published articles. (**D**) HLA-I binding prediction for immunopeptides identified in HT29-CTRL and HT29-BIR, as performed by netMHCpan4.1. “Total” includes peptides that bind to at least one HLA-I allele. (**E**) HLA-I binding prediction for immunopeptides identified in MCF7-CTRL and MCF7-BIR, as performed by netMHCpan4.1. “Total” includes peptides that bind to at least one HLA-I allele. **: *p* < 0.005.

**Figure 3 ijms-25-03660-f003:**
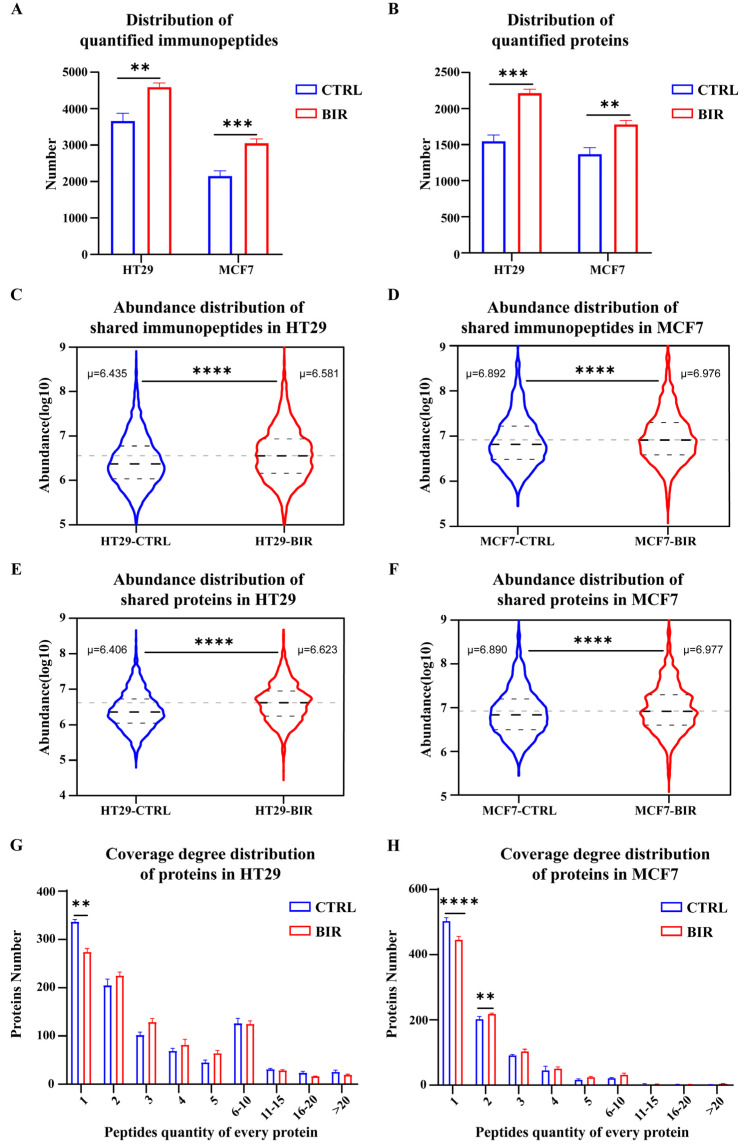
Label-free quantitation (LFQ) of the immunopeptidome and source proteins. (**A**) In HT29 cells, quantitated immunopeptides were 1.25-fold increased (from 3669 to 4594) after Birinapant treatment. In MCF7 cells, quantitated immunopeptides were 1.42-fold increased (from 2155 to 3054) after Birinapant treatment. (**B**) Several different quantitated immunopeptides may be derived from the same protein; therefore, quantitated proteins more accurately reflect antigen presentation diversity. In HT29 cells, quantitated proteins were 1.43-fold increased (from 1549 to 2215); and in MCF7 cells, quantitated proteins were 1.30-fold increased (from 1370 to 1782). Abundance of shared immunopeptides following logarithm transformation between BIR and CTRL in (**C**) HT29 cells and (**D**) MCF7 cells was significantly increased after Birinapant treatment. Abundance of shared proteins between BIR and CTRL in (**E**) HT29 cells and (**F**) MCF7 cells was also increased. For each protein, the identified peptides reflect the degree of coverage. The number of proteins that presented a different quantity of peptides (from 1 to >20) in HT29 cells (**G**) and MCF7 cells (**H**). The number of proteins presenting only one peptide decreased after Birinapant treatment. Bold dotted line in each violin of (**C**–**F**) means average abundance, while other two dotted lines mean quartile. **: *p* < 0.005; ***: *p* < 0.0005; ****: *p* < 0.0001.

**Figure 4 ijms-25-03660-f004:**
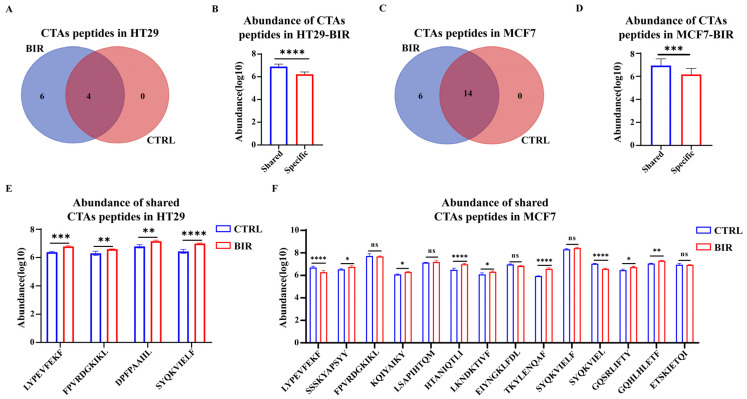
Cancer/testis antigen (CTA) peptides identified in the immunopeptidome. CTAs are proteins expressed in testis or cancer. (**A**) In HT29 cells, 10 CTA peptides were identified in BIR, including 4 that were also identified in CTRL. (**B**) The 4 shared CTA peptides had a higher abundance in comparison with the 6 specific CTA peptides in BIR. (**C**) In MCF7 cells, a total of 14 CTA peptides were identified in CTRL, all of which were covered in the 20 identified in BIR. (**D**) The 14 shared CTA peptides had a higher abundance in comparison with the 6 specific CTA peptides in BIR. (**E**) In HT29, the 4 shared CTA peptides had a higher abundance in BIR. Data are averaged from triplicate samples. (**F**) Among the 14 shared CTA peptides in MCF7 cells, 7 had a significantly higher abundance in BIR, 5 showed no obvious difference (still 2 peptides had a higher abundance), and only 2 peptides had a lower abundance in BIR. Sequences and other information of CTA peptides are listed in Table 1 (HT29) and Table 2 (MCF7). *: *p* < 0.05; **: *p* < 0.005; ***: *p* < 0.0005; ****: *p* < 0.0001.

**Figure 5 ijms-25-03660-f005:**
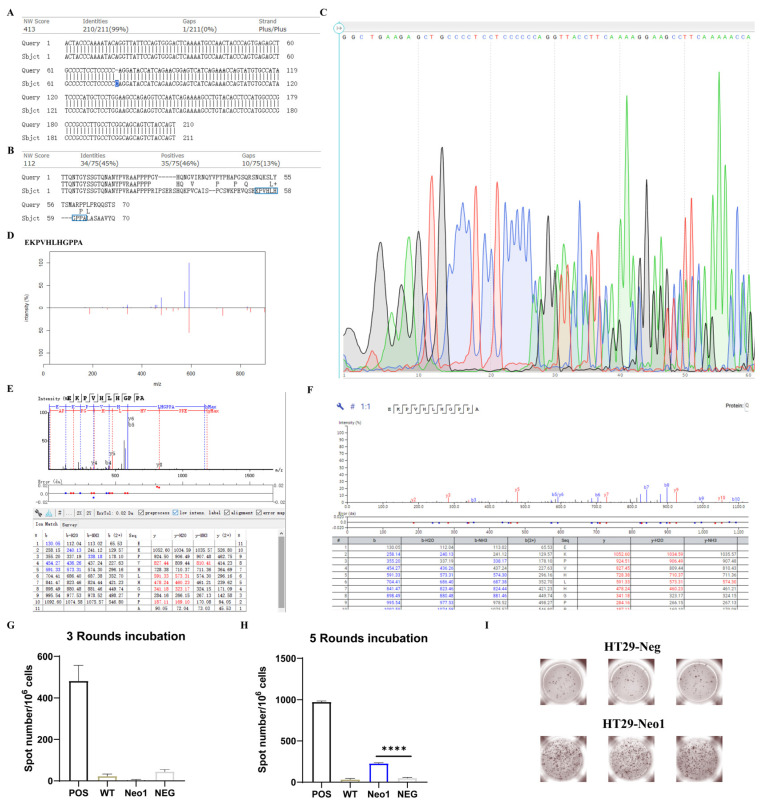
Validation of neoantigen existence and immunogenicity. (**A**) Blast between part of the MYO1E exon nucleotide of wide-type and Indel. Sequences were excerpted from 2802 to 3012 of MYO1E exon nucleotide. The upper is wide-type, and the bottom is Indel. The blue marked “C” is insertion mutation. (**B**) Blast between parts of the MYO1E protein sequence of wide-type and Indel. Sequences were excerpted from 934 to 1004 of MYO1E protein sequence. The upper is wide-type, and the bottom is Indel. The blue marked peptide is Neo1. (**C**) Part of the sequencing results of MYO1E in HT29 mRNA. (**D**) Spectrum comparison of neo1. Upper is presented peptide and bottom is synthetic peptide. (**E**) MS2 of the presented Neo1. (**F**) MS2 of the synthesized Neo1. (**G**) After three rounds of pre-stimulation, Neo1 failed to show further IFN-γ spots. (**H**) To avoid false negative results caused by inadequate stimulation, 5 rounds of pre-stimulation were performed. Neo1 showed a significant increase in IFN-γ spots. (**I**) Spot distributions of Neo1 are shown. ****: *p* < 0.0001.

**Figure 6 ijms-25-03660-f006:**
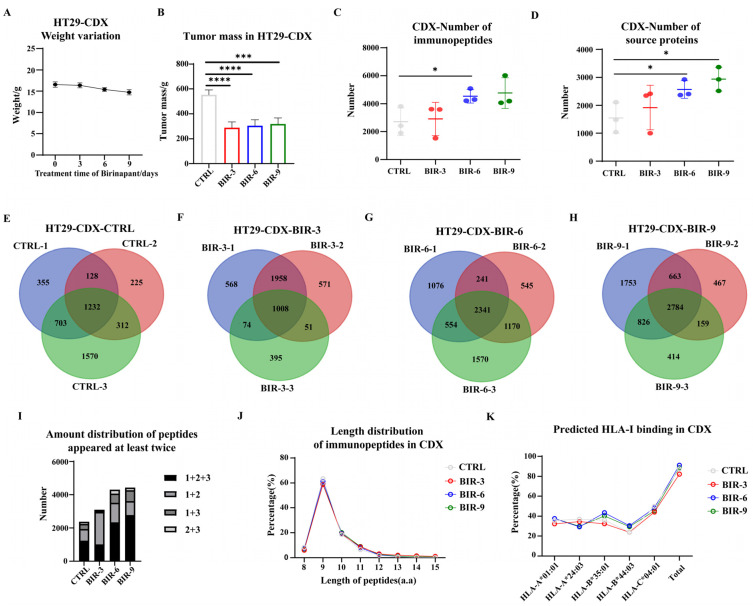
Basic characteristics of the immunopeptidome in HT29 cell-derived xenografts (CDXs). (**A**) Weight variation of HT29 cell-derived xenografts (CDXs) after Birinapant treatment. (**B**) Tumor mass variation of HT29-CDX after Birinapant treatment. CTRL were administered saline for 6 days, and BIR-3, BIR-6, and BIR-9 were administered Birinapant for 3 days, 6 days, and 9 days, respectively. (**C**) Triplicate immunopeptides in each group identified 2711, 2908, 4531, and 4844 peptides, respectively. The number of peptides increased with the prolonged Birinapant treatment time. (**D**) For the source proteins of immunopeptides, 1544, 1919, 2564, and 2938 proteins were identified in the four groups, respectively, also indicating a positive correlation with Birinapant treatment time. Overlap between triplicate samples of HT29-CDX-CTRL (**E**), HT29-CDX-BIR-3 (**F**), HT29-CDX-BIR-6 (**G**), and HT29-CDX-BIR-9 (**H**). (**E**) Specifically, 2418, 1897, and 3817 peptides were identified in HT29-CDX-CTRL; 1232 were shared, and 2375 were identified at least twice. (**F**) A total of 3608, 3588, and 1528 peptides were identified in HT29-CDX-BIR-3; 1008 were shared, and 3097 were identified at least twice. (**G**) A total of 4212, 4297, and 5083 peptides were identified in HT29-CDX-BIR-6; 2341 were shared, and 4306 were identified at least twice. (**H**) A total of 6026, 4073, and 4183 peptides were identified in HT29-CDX-BIR-9; 2784 were shared, and 4432 were identified at least twice. (**I**) For peptides identified at least twice, the specific number of peptides belonging to “1 + 2 + 3”, “1 + 2”, “1 + 3”, or “2 + 3” are listed. (**J**) Length distribution of immunopeptides in CDX. There was little difference between the four groups. (**K**) HLA-I binding affinity as predicted by netMHCpan4.1. For each HLA-I allele, although the trend of change in each group was not entirely consistent, the differences between groups were not significant. *: *p* < 0.05; ***: *p* < 0.0005; ****: *p* < 0.0001.

**Figure 7 ijms-25-03660-f007:**
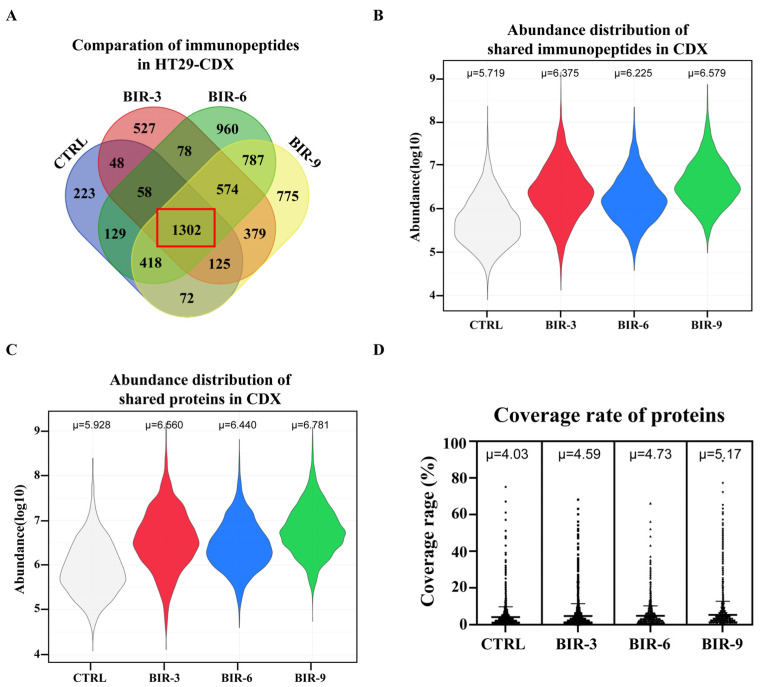
Abundance distribution and functional clustering in the immunopeptidome. (**A**) Comparison of immunopeptides in HT29-CDX: 1302 peptides (inside the red box) were shared among the four groups. (**B**) Label-free quantitation (LFQ) was used to obtain the abundance of immunopeptides. Average abundance by logarithm transformation was calculated for immunopeptides shared by all four groups (1302 peptides; see Appendix A). The μ value in BIR-6 (6.225) was lower than that in BIR-3 (6.375), indicating that the abundance distribution does not exhibit a positive correlation with Birinapant treatment time. (**C**) For source proteins showing a consistent tendency with peptides, the μ value in BIR-6 (6.440) was lower than that in BIR-3 (6.560). (**D**) Comparison of the coverage rate of proteins among the four groups. The average coverage rate was 4.03, 4.59, 4.73, and 5.17 in CTRL, BIR-3, BIR-6, and BIR-9, respectively.

**Figure 8 ijms-25-03660-f008:**
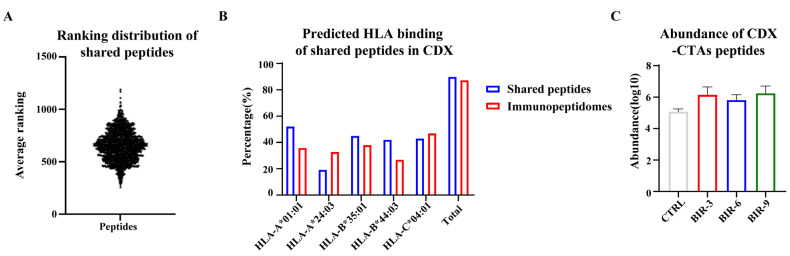
Characteristics of shared and CDX-CTA peptides. (**A**) Ranking distribution of the 1302 shared immunopeptides according to the average abundance in each group: BIR-3/CTRL, BIR-6/BIR-3, and BIR-9/BIR-6. The three parameters were ranked separately to obtain the average ranking for each peptide. No peptides showed an absolute top ranking, nor did any have an average ranking lower than 200, indicating that Birinapant may influence antigen presentation in specific pathways rather than specific proteins or peptides. (**B**) HLA-I binding prediction of 1302 shared peptides. Data were calculated based on the binding prediction for each group (Figure 7D). (**C**) Abundance distribution of all CTA peptides. The average abundance was 5.08, 6.14, 5.81, and 6.24 in CTRL, BIR-3, BIR-6, and BIR-9, respectively.

**Table 1 ijms-25-03660-t001:** CTA peptides identified in HT29-BIR and HT29-CTRL.

Sequence	Group	Length	Immunogenicity	Binding Affinity (nM)	CTA
DPFPAAIIL	BIR and CTRL	9	0.29185	330.78	PBK
FAITEPLVTF	BIR	10	0.21604	83.05	CEP55
VYVPHIHVW	BIR	9	0.21044	3.93	ATAD2
LYPEVFEKF	BIR and CTRL	9	0.20605	2.46	ATAD2
SALPTVVAY	BIR	9	0.1196	6.14	SPAG1
LLDDTGLAY	BIR	9	0.11272	11.13	CNOT9
TAAPVPTTL	BIR	9	0.08674	320.51	CNOT9
SYQKVIELF	BIR and CTRL	9	−0.0111	1.92	PBK
FPVRDGKIKL	BIR and CTRL	10	−0.0966	714.63	KDM5B
FLDPRQPSY	BIR	9	−0.16862	95.31	DCAF12

Note: the immunogenicity was stronger at a peptide score greater than 0.1; these are marked in red.

**Table 2 ijms-25-03660-t002:** CTA peptides identified in MCF7-BIR and MCF7-CTRL.

Sequence	Group	Length	Immunogenicity	Binding Affinity (nM)	CTA
**KKYAEDRERFF**	BIR	11	0.414	26,765.26	KIF20B
**AEDRERFF**	BIR	8	0.3043	4319.36	KIF20B
DPFPAAIIL	BIR	9	0.29185	7475.95	PBK
GQSRLIFTY	BIR and CTRL	9	0.23434	783.95	KIF20B
LYPEVFEKF	BIR and CTRL	9	0.20605	13,409.88	ATAD2
GQHLHLETF	BIR and CTRL	9	0.12758	4437.84	PRAME
LSAPIHTQM	BIR and CTRL	9	0.12667	1147.05	KNL1
HTANIQTLI	BIR and CTRL	9	0.05303	3629.06	KNL1
**KQIYAIKY**	BIR and CTRL	8	0.02086	9150.76	TTK
TKYLENQAF	BIR and CTRL	9	0.00415	4136.18	KIF2C
SYQKVIEL	BIR and CTRL	8	−0.00462	25,404.92	PBK
SYQKVIELF	BIR and CTRL	9	−0.0111	6746.93	PBK
**LKNDKTIVF**	BIR and CTRL	9	−0.0168	8977.57	KNL1
**ETSKIETQI**	BIR	9	−0.08177	23,727.05	KIF20B
EIYNGKLFDL	BIR and CTRL	10	−0.08339	29,211.47	KIF2C
FPVRDGKIKL	BIR and CTRL	10	−0.0966	24,543.03	KDM5B
**SKYAPSYY**	BIR	8	−0.13148	28,744	ATAD2
TKIATKMGF	BIR	9	−0.21103	17,176.42	KDM5B
ETASAMATL	BIR and CTRL	9	−0.22527	24,710.9	KDM5B
**SSSKYAPSYY**	BIR and CTRL	10	−0.38842	15,093.97	ATAD2

Note: peptides that have not previously been identified in the IEDB database are shown in bold. The immunogenicity was stronger at a peptide score greater than 0.1; these are marked in red.

**Table 3 ijms-25-03660-t003:** Neoantigens identified in cell lines.

Sequence	Group	Protein	−10LgP	Type	Binding Affinity (nM)	Binding Percentage (%)
LPIIQK**V**EPQ	HT29 BIR	PLIN2 L72V	9.26	SNV	30,867.68	2.2(B*35:01)
FTP**V**EEFVP	HT29 CTRL	HAP1 A480V	5.88	SNV	26,466.92	1.4(B*35:01)
KLSP**Y**LAR	HT29 BIR	DNHD1 H2861Y	6.73	SNV	19,405.62	32(A*01:01)
EV**L**LQLPT	HT29 BIR	C1orf87 P228L	7.88	SNV	19,127.34	44(A*01:01)
**EKPVHLHGPPA** (Neo1)	HT29 BIR	MYO1E	6.95	Indel	35,015.57	0.45(B*35:01)
PGPPLIPVPVG**V**	MCF7 BIR and CTRL	DNM2 A796V	6.66	SNV	1058.45	0.12(A*02:01)
RIQRAYKLY**R**	MCF7 BIR	ASPM L3132R	5.7	SNV	24,357.58	1.6(C*05:01)
**L**YLTAETLKNRM	MCF7 CTRL	NPIPB6 P232L	6.77	SNV	20,515.42	0.63(B*44:02)
DGANRHIT**N**	MCF7 BIR	CLPB S187N	7.07	SNV	33,381.86	23(B*18:01)
GPISVPIPGPI**S**	MCF7 BIR	TPRX1 P204S	7.54	SNV	35,104.33	3.8(C*05:01)

Note: the mutational positions are shown in bold. Peptides belonged to Indel derived from frameshifts; thus, the whole sequence was mutated. SNV: single-nucleotide variant; Indel: insertion/deletion mutation. The alleles in binding percentage were the most likely alleles for the peptide to bind. Neo1 were chose for immunogenicity assessment.

**Table 4 ijms-25-03660-t004:** CTA peptides identified in HT29-CDX.

Sequence	Group	Length	Immunogenicity	Binding Affinity (nM)	CTA
DPFPAAIIL	BIR-3, BIR-6, and BIR-9	9	0.29185	330.78	PBK
AYAIIKEEL	BIR-3 and BIR-9	9	0.21622	27.20	ATAD2
VYVPHIHVW	BIR-3, BIR-6, and BIR-9	9	0.21044	3.93	ATAD2
LYPEVFEKF	CTRL, BIR-3, BIR-6, and BIR-9	9	0.20605	2.46	ATAD2
SALPTVVAY	BIR-6 and BIR-9	9	0.1196	6.14	SPAG1
LLDDTGLAY	BIR-9	9	0.11272	11.13	CNOT9
TAAPVPTTL	CTRL, BIR-6, and BIR-9	9	0.08674	320.51	CNOT9
HANDQTVIF	BIR-6	9	0.05656	12.86	KNL1
IATSHNIVY	BIR-6	9	0.00798	11.59	KNL1
SYQKVIELF	BIR-3, BIR-6, and BIR-9	9	−0.0111	1.92	PBK
SFNEAMTQI	CTRL and BIR-6	9	−0.06347	149.40	KIF2C
EITGMNTL	BIR-6	8	−0.09763	19,865.35	KNL1
DEAVGVQKW	BIR-6 and BIR-9	9	−0.09766	67.79	BLTP2
KYAPSYYHV	BIR-3	9	−0.14726	3.47	ATAD2
FLDPRQPSY	BIR-6 and BIR-9	9	−0.16862	26.67	DCAF12
ETEESNLNMY	BIR-3	10	−0.15265	33.81	ATAD2
EENQKRYYL	BIR-3, BIR-6, and BIR-9	9	−0.28522	720.96	ATAD2
RYSGVNQSMLF	BIR-3	11	−0.40891	3.18	ATAD2

Note: the immunogenicity was stronger at a peptide score greater than 0.1; these are marked in red.

**Table 5 ijms-25-03660-t005:** Neoantigens identified in HT29-CDX.

Sequence	Group	Protein	−10LgP	Type	Immuno-Genicity	Binding Affinity (nM)	Binding Percentage
LRVQ**L**HLKK	BIR-3	DNAH3 P1197L	22.69	SNV	−0.21887	35,502.71	17.805
LRVQPHLKK	—	DNAH3 WT	—	WT	—	38,195.54	14.797
SEIRHTA**N**RW	BIR-6 and BIR-9	HSPB1 D93N	24.99	SNV	0.22619	11.18	0.0099
SEIRHTADRW	—	HSPB1 WT	—	WT	—	13.98	0.03

Note: the mutational positions are shown in bold. SNV: single-nucleotide variant; WT: wide-type.

## Data Availability

The mass spectrometry data have been deposited to the ProteomeXchange Consortium (http://proteomecentral.proteomexchange.org, accessed on 31 August 2023) via the iProX partner repository, with the dataset identifiers PDX045302, PDX045306, and PDX045308. The sequencing data have been deposited in the NCBI Sequence Read Archive (SRA) database, the submission number is SRR27068101 and SRR27068102.

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
