# Peer review of "Birinapant Reshapes the Tumor Immunopeptidome and Enhances Antigen Presentation"

_ijms, 2024, doi:10.3390/ijms25073660_

Round 1
Reviewer 1 Report
Comments and Suggestions for Authors
This is a well written paper examining the effects of of Birinapant an agent that alters apoptosis in tumors and shows some significant therapeutic efficacy in a variety of tumors may also enhance antigen presentation and reshape the tumor immunopeptidome. It apparently does so by stimulating the presentation of potential peptides associated with the MHC Class I complex of proteins expressed on various nucleated cells. Thus it both increases the number of peptides expressed as well as altering the classes of potential immunopeptides expressed.
The paper clearly presents the relatively complex methodologies and the results obtained in the paper. However as discussed by the authors there are certain practical limitations to the approach which make it a bit more difficult for most investigators to fully use/appreciate the results. There are a huge number of immunopeptides identified thus determining which are relevant to tumor rejection is a major hurdle.
The reviewer things that if the authors might briefly discuss potential use of Birinapant in the context of a defined vaccines that might be of some use. Thus in an animal model where animals are vaccinated against defined antigens would administration of Birinapant administration potentially increase levels of the relevant immunopeptides and make the tumors more susceptible to the vaccine itself. Second might use of Birinapant enhance the initial vaccination process as wll.
Author Response
Answer:
Thanks for your comment. This is a great advice. We actually also believe that Birinapant may serve as a vaccine adjuvant and play a role in some treatments. We have added sentences to the Discussion section to make this statement. Thank you for your suggestion. The details are as follows:
We believed that Birinapant has the potential to be used as an adjuvant, enhance the immunogenicity of tumor cells and make patients more benefit from immunotherapy. Currently, as the most effective immunotherapy, ICB still has limited efficacy in some tumors (such as glioblastoma and pancreatic cancer). Gu et al have demonstrated that Birinapant combined with ICB can further inhibit tumor growth, which suggested a possible strategy to address poor performance of ICB in some cases. On the other hand, Birinapant enhances the antigen presentation pathway of tumor cells. For CAR-T or TCR-T therapy, this may lead better recognition. Furthermore, for neoantigens vaccines and dendritic cell vaccines, they may also benefit from better antigen presentation by tumor cells.

Reviewer 2 Report
Comments and Suggestions for Authors
This is a potentially interesting manuscript though many concerns here though these concerns are intrinsic to this ambitious approach.
The contribution by Zhang et al. tried to make the case for the SMAC mimetic Birinapant in prospective immunotherapy, in view of a recent paper (cited as Ref. 39). The authors used a standard immunopeptidomics platform to report that the chemical upregulates a total complement of peptides associated with HLA-I in terms of diversity and abundance. They went on to identify what they claim to be immunogenic neoantigens. Lastly, they looked at the ability of Birinapant to modulate an immunopeptide landscape using a mouse model bearing HT29 cells.
1. The most serious concern is to what extent the identified peptides should be trusted, especially neoantigens. That said, "some" of the neoantigens (Fig 6) appear to be immunogenic, more or less. In the Abstract part, the authors stated the immunogenicity of “four” neoantigens. Please make it consistent and unambiguous on the numbers.
The investigators in this field are always anxious about the robustness on the mass spectrometry identification (Fig S6) here, for the seven neoantigens.
The authors are strongly advised to show MS/MS spectra for each synthetic peptide (they already have ones used for ELISpot) to make sure the spectra are correctly interpreted. Along this line, they stated (Line 302) that two neoantigens (neo 3, EKPVHLHGPPA and neo 7, GPISVPIPGPIS) are derived from indels. This is very interesting, but as they did note, that sequences of this kind are quite exceptional in immunopeptidomics. More concrete data presentation help the readers to believe their claim, for instance, a figure depicting the matches between a given “identified” sequence and its theoretical one deduced from de novo or data-based searches accompanied by those from the synthetic peptides. The identification of these neoantigens constitutes an essential part of this work, so mass spectrometry annotation data should be brought to the main text (at least for neo 3 and 7), too, rather than being put away in Fig S6.
As for Fig S6, the figures were generated by the software PEAKS, so the authors should be able to provide product ion annotations (b-, y-ions, or c-, z-ions) and the corresponding sequence for each sequence without undue difficulty. Put "Query" on the upper part and "Predicted" or "Library" on the bottom. The same must go for the seven synthetic peptides. The current presentation doesn’t guarantee that those sequences were actually identified, whatever the algorithm the authors may have adopted. If the authors are quite sure about their interpretation, they might as well put mass spectrometry raw data in a public repository, like PRIDE Archive, MassIVE, jPOST or iProX.
We know, the grim truth is that peptidomics papers, even those published in high profile journals, abound with innumerable erroneous sequences, because of lax or poor spectrum interpretation. Thus please be careful about the possible incompleteness in interpretation and arguments.
2. To construct a cell line-specific database, they conducted WES, but didn’t mention RNAseq, which this reviewer feel was unlikely to be the norm. Was the WES info alone sufficient to create a reliable database for neoantigen discovery?
3. In the Abstract part, no real finding has been given about GO analysis, reflecting the lack of experiments underpinning the GO analysis data (Figs 4, 8, 9). So, the authors are advised to transfer the GO terms figures to the Supplement to stress the major points they have to say in this paper. Instead, they might want to bring Fig S7 (ELISpot) and Fig S8 (transplanted HT29 immunopeptidome) to the main body, which would definitely improve the impact of this paper, besides the mass spectrometry issue raised above. Also, passages describing the GO analysis (e.g., Lines 197 to 223) should be cut down to allow for Figs S7 and S8 (mass spectrometry annotation figures as well) coming into the main text.
4. Put together Tables 3 and 4 as they highly overlap.
5. Lines 244-245, the authors tried to state that the machinery for antigen processing/presentation is upregulated by BIR, but the data presented is too tenuous to be persuasive. TAP consists of TAP-1 and -2, and PSMB has multiple subunit types. As per Fig S5, the authors appeared to quantitate TAP and PSMB with TMT-based quantitative mass spectrometry, so they should be able to name each component and its ratio (e.g., PSMB type1, type 2, etc.). Besides, did the authors look at some of them by immunoblotting?
6. Describe the control peptide’s identity used for the ELISpot assay, as well as provide the number of technical replicates.
7. Did the authors test the Table 6 neoantigens on an ELISpot assay?
8. The authors stated that a combination of ELISpot and FCM to reliably verify the immunogenicity of neoantigens (Line 554), but where can we find the FCM data to support this? It seems that they used FCM only to examine HLA-I expression levels on different cell lines treated with Birinapant (Fig S1).
9. In the Methods section, immunoblotting is mentioned, but no data has been provided.
Minor
Fig. 1 is well-illustrated for a general audience. Letters like “in silico prediction” would make it better.
Line 127: Insert the reference number of 39 (by Gu et al.) right after HLA.
Line 260: The passage followed by the conjunctive adverb "therefore" sounds a bit uncomfortable. “We attempted to detect” or something else would be fine.
Line 305: "varied from 0.12 to 44" needs a unit of measure to make it readable. Looks like the figures are from Table 3.
Lines 305 to 308: The passage sounds unconvincing, even though Ref 48 has been cited. Please modify it so the message can be more clearly conveyed to readers.
Lines 346 to 348: The passage has multiple grammar errors.
Line 430: “peptide-derived proteins” is not a correct language. The authors did TMT proteomics. Did they see some precursors (=parent proteins) from which the identified CTA peptides or neoantigens are supposed to be processed out? Please bring them up in the main text if they did.
Line 619: generate instead of generation
Line 638: "an amplitude of 70" doesn't make sense unless the ultrasonicator used (manufacturer, product type) is provided.
Line 659: They might want to try with the latest version of PEAKS Studio 11 to excavate more sequences, rather than Studio X Pro they currently used. They should describe the nature of a library for conducting DIA, in silico or one made after DDA studies?
Line 660: DIA refers to NOT database-independent, but data-independent.
Line 661: Describe the grounds for using iodoacetamide (carbamidomethylation mentioned) in the immunopeptidomics part. In the current version, there seems to be a mix-up between peptidomics and proteomics procedures.
Line 664: NOT P.P.M. but ppm.
Line 671: The figure 0E0?
Line 681: Delete the definite article and rewrite as appropriate.
Line 699: Spell out HRP. Obviously, HRP in Line 724 is horseradish peroxidase, which still remains undefined elsewhere.
Line 727: Describe the identity of the positive stimulant.
Line 748: The verb administer needs an object as long as “mice” is the subject.
At the risk of sounding pedagogic, the authors are advised not to start a sentence with a numeral (to name a few, 200 uL W6/32-conjugated....).
It is assumed that figures with a higher resolution are going to appear in the event of its publication. Please be advised that a dash is longer than a hyphen and they are easy to mix up. Dashes should be used sparingly in figure titles; dashes look like hyphens (e.g., HT29-predicted, CDX-predicted). Fig. 5, “comparation” doesn’t sound natural.
Fig S3 and S9 legends, each allele sounds natural. Add predicted ahead of binding affinity.
Comments on the Quality of English Language
Readable, but check the several parts (I am not native speaker).
Author Response
Q: 1. The most serious concern is to what extent the identified peptides should be trusted, especially neoantigens. That said, "some" of the neoantigens (Fig 6) appear to be immunogenic, more or less. In the Abstract part, the authors stated the immunogenicity of “four” neoantigens. Please make it consistent and unambiguous on the numbers.
The investigators in this field are always anxious about the robustness on the mass spectrometry identification (Fig S6) here, for the seven neoantigens.
The authors are strongly advised to show MS/MS spectra for each synthetic peptide (they already have ones used for ELISpot) to make sure the spectra are correctly interpreted. Along this line, they stated (Line 302) that two neoantigens (neo 3, EKPVHLHGPPA and neo 7, GPISVPIPGPIS) are derived from indels. This is very interesting, but as they did note, that sequences of this kind are quite exceptional in immunopeptidomics. More concrete data presentation help the readers to believe their claim, for instance, a figure depicting the matches between a given “identified” sequence and its theoretical one deduced from de novo or data-based searches accompanied by those from the synthetic peptides. The identification of these neoantigens constitutes an essential part of this work, so mass spectrometry annotation data should be brought to the main text (at least for neo 3 and 7), too, rather than being put away in Fig S6. As for Fig S6, the figures were generated by the software PEAKS, so the authors should be able to provide product ion annotations (b-, y-ions, or c-, z-ions) and the corresponding sequence for each sequence without undue difficulty. Put "Query" on the upper part and "Predicted" or "Library" on the bottom. The same must go for the seven synthetic peptides. The current presentation doesn’t guarantee that those sequences were actually identified, whatever the algorithm the authors may have adopted. If the authors are quite sure about their interpretation, they might as well put mass spectrometry raw data in a public repository, like PRIDE Archive, MassIVE, jPOST or iProX.
As for Fig S6, the figures were generated by the software PEAKS, so the authors should be able to provide product ion annotations (b-, y-ions, or c-, z-ions) and the corresponding sequence for each sequence without undue difficulty. Put "Query" on the upper part and "Predicted" or "Library" on the bottom. The same must go for the seven synthetic peptides. The current presentation doesn’t guarantee that those sequences were actually identified, whatever the algorithm the authors may have adopted. If the authors are quite sure about their interpretation, they might as well put mass spectrometry raw data in a public repository, like PRIDE Archive, MassIVE, jPOST or iProX.
We know, the grim truth is that peptidomics papers, even those published in high profile journals, abound with innumerable erroneous sequences, because of lax or poor spectrum interpretation. Thus please be careful about the possible incompleteness in interpretation and arguments.
Answer: Thanks for this comment. We take this consideration seriously and have revised some of our results based on your suggestion.
As you said, current immunopeptidomics papers mainly rely on synthetic peptide spectral comparison to verify the reliability of identified peptides, which actually relies more on subjective evaluation and may therefore sometimes lead to erroneous sequences. We think that instead of getting more results, it may be better to demonstrate a reliable result. In order to avoid this problem as much as possible in our work, we conducted a more stringent screening of the ten identified neoantigens and selected the best - Neo1 (EKPVHLHGPPA) - which we believe has a good MS2 and good binding percentage. Subsequently, to further demonstrate the existence of this neoantigen, we extracted the mRNA from the HT29 cell line and sequenced it. The results showed that both the WT and mutant forms of MYO1E appeared in the transcript. Subsequently, we synthesized this peptide and performed spectral comparison, which also showed a good match. The above results will be added to the main text as F5 together with the immunogenicity validation of Neo1.
In addition, the MS2 of synthesized Neo1 and presented Neo1 will be placed in F5 for reference.
- To construct a cell line-specific database, they conducted WES, but didn’t mention RNAseq, which this reviewer feel was unlikely to be the norm. Was the WES info alone sufficient to create a reliable database for neoantigen discovery?
Answer: Thanks for this comment. Although many current immunopeptidome databases were constructed based on RNA-seq, there are still some database constructions based on WES. This is mainly considered from different perspectives, RNA-seq is from the transcriptome level, and WES is from the genome level.
(reference: 30; Mona O Mohsen et al., 2022; Rui Chen et al., 2019.)
- In the Abstract part, no real finding has been given about GO analysis, reflecting the lack of experiments underpinning the GO analysis data (Figs 4, 8, 9). So, the authors are advised to transfer the GO terms figures to the Supplement to stress the major points they have to say in this paper. Instead, they might want to bring Fig S7 (ELISpot) and Fig S8 (transplanted HT29 immunopeptidome) to the main body, which would definitely improve the impact of this paper, besides the mass spectrometry issue raised above. Also, passages describing the GO analysis (e.g., Lines 197 to 223) should be cut down to allow for Figs S7 and S8 (mass spectrometry annotation figures as well) coming into the main text.
Answer: Thanks for this comment. We've modified based on your comments. The details are as follows: We modified the text in parts 197-223 to make it more concise, and adjusted F4 to SF5; the original SF7 and SF8 were added to the text, and SF7 and F5 were merged to fully express the Elispot results. SF8 merged with F6. The C part of F8 is added to SF8. The D part of S9 is added to SF9.
- Put together Tables 3 and 4 as they highly overlap.
Answer: Thanks for this comment. We have put them together.
- Lines 244-245, the authors tried to state that the machinery for antigen processing/presentation is upregulated by BIR, but the data presented is too tenuous to be persuasive. TAP consists of TAP-1 and -2, and PSMB has multiple subunit types. As per Fig S5, the authors appeared to quantitate TAP and PSMB with TMT-based quantitative mass spectrometry, so they should be able to name each component and its ratio (e.g., PSMB type1, type 2, etc.). Besides, did the authors look at some of them by immunoblotting?
Answer: Thanks for this comment. In fact, we performed statistical analyzes on different proteins of the same type to draw easy-to-understand conclusions. We list here the raw data from the mass spectrometry results for reference. At the same time, we have also verified some proteins through WB and QPCR, which are also listed here for reference.
This figure showed the fold change of proteins in proteomics. We could find many HLA, TAP and PSMB proteins up-regulated after Birinapant treatment (fold change >1).
This figure showed the fold change of genes by QPCR.
This figure showed the change of proteins by WB. ASTX means ASTX660, XEV means Xevinapant, both of them belong to SMAC mimetics.
- Describe the control peptide’s identity used for the ELISpot assay, as well as provide the number of technical replicates.
Answer: Thanks for this comment. For Indels, since wild-type peptide was not present, they were compared to negative controls. All technical replicates were three times.
- Did the authors test the Table 6 neoantigens on an ELISpot assay?
Answer: Thanks for this comment. The neoantigens in Table 6 were not tested. This is limited by the amount of PBMCs. PBMCs were purchased from commercial companies, and we failed to obtain enough number of PBMCs to perform our ELISPOT verification of subsequent neoantigens identified from CDX.
- The authors stated that a combination of ELISpot and FCM to reliably verify the immunogenicity of neoantigens (Line 554), but where can we find the FCM data to support this? It seems that they used FCM only to examine HLA-I expression levels on different cell lines treated with Birinapant (Fig S1).
Answer: Thanks for this comment. Sorry, this was a mistake and we have corrected it in the manuscript.
- In the Methods section, immunoblotting is mentioned, but no data has been provided.
Answer: Thanks for this comment. We used immunoblotting in the experiments to screen elution components containing HLA during immunoprecipitation, so it was written in the Methods section. The specific experimental results could be seen in the “western blot pictures”.
Minor:
Fig. 1 is well-illustrated for a general audience. Letters like “in silico prediction” would make it better.
Answer: Thanks for this comment. I am sorry that we don’t use in silico prediction here, so I don’t quite understand which part you are mentioned. Could you please describe it in more detail? Thank you for your patience very much.
Line 127: Insert the reference number of 39 (by Gu et al.) right after HLA.
Answer: Thanks for this comment. I finished this question.
Line 260: The passage followed by the conjunctive adverb "therefore" sounds a bit uncomfortable. “We attempted to detect” or something else would be fine.
Answer: Thanks for this comment. I finished this question.
Line 305: "varied from 0.12 to 44" needs a unit of measure to make it readable. Looks like the figures are from Table 3.
Answer: Thanks for this comment. I modified this sentence.
Lines 305 to 308: The passage sounds unconvincing, even though Ref 48 has been cited. Please modify it so the message can be more clearly conveyed to readers.
Answer: Thanks for this comment. I modified this sentence.
Lines 346 to 348: The passage has multiple grammar errors.
Answer: Thanks for this comment. I modified this sentence.
Line 430: “peptide-derived proteins” is not a correct language. The authors did TMT proteomics. Did they see some precursors (=parent proteins) from which the identified CTA peptides or neoantigens are supposed to be processed out? Please bring them up in the main text if they did.
Answer: Thanks for this comment. I modified this sentence. Although we identified many precursors of CTAs or neoantigens in proteomics, we did not identify peptides with exact sequence matches.
Line 619: generate instead of generation
Answer: Thanks for this comment. I modified this sentence.
Line 638: "an amplitude of 70" doesn't make sense unless the ultrasonicator used (manufacturer, product type) is provided.
Answer: Thanks for this comment. I deleted this information.
Line 659: They might want to try with the latest version of PEAKS Studio 11 to excavate more sequences, rather than Studio X Pro they currently used. They should describe the nature of a library for conducting DIA, in silico or one made after DDA studies?
Answer: Thanks for this comment. Different versions of PEAKS may lead to subtle differences in peptide results, but we believe this has a small impact on drawing our conclusions. We have used Proteome Discovery 2.2 to search the same raw data. Although the number of identified peptides is different, the feature analysis is consistent. Our DDA library is formed by integrating the human reviewed protein database of UniProt and the mutation protein database obtained in WES. For DIA, we used the DENOVO method provided by PEAKS.
Line 660: DIA refers to NOT database-independent, but data-independent.
Answer: Thanks for this comment. I modified this sentence.
Line 661: Describe the grounds for using iodoacetamide (carbamidomethylation mentioned) in the immunopeptidomics part. In the current version, there seems to be a mix-up between peptidomics and proteomics procedures.
Answer: Thanks for this comment. We referred to published literature and added IAA in our method. In fact, IAA is added in both immunopeptidomics and proteomics experiments.
Line 664: NOT P.P.M. but ppm.
Answer: Thanks for this comment. I modified this sentence.
Line 671: The figure 0E0?
Answer: Thanks for this comment. This is a mistake, I modified this sentence.
Line 681: Delete the definite article and rewrite as appropriate.
Answer: Thanks for this comment. I modified this sentence.
Line 699: Spell out HRP. Obviously, HRP in Line 724 is horseradish peroxidase, which still remains undefined elsewhere.
Answer: Thanks for this comment. I modified this sentence.
Line 727: Describe the identity of the positive stimulant.
Answer: Thanks for this comment. I added this information. Positive stimulant is phorbol myristoyl acetate (PMA)
Line 748: The verb administer needs an object as long as “mice” is the subject.
Answer: Thanks for this comment. I modified this sentence.
At the risk of sounding pedagogic, the authors are advised not to start a sentence with a numeral (to name a few, 200 uL W6/32-conjugated....).
Answer: Thanks for this comment. I modified some sentence.
It is assumed that figures with a higher resolution are going to appear in the event of its publication. Please be advised that a dash is longer than a hyphen and they are easy to mix up. Dashes should be used sparingly in figure titles; dashes look like hyphens (e.g., HT29-predicted, CDX-predicted). Fig. 5, “comparation” doesn’t sound natural.
Answer: Thanks for this comment. I modified some figures.
Fig S3 and S9 legends, each allele sounds natural. Add predicted ahead of binding affinity.
Answer: Thanks for this comment. I modified this sentence.

Reviewer 3 Report
Comments and Suggestions for Authors
1. Please expand on current understanding of birinapant's effect on cancer cells both in vitro and in vivo
2. Why did the authors choose MCF7 cell line since we know TNBC is not a highly immunogenic cell line
3. While DMSO is used as a vehicle control (necessary since it is the vehicle for birinapant) more accurate comparison of birinapant's effect on in vitro immunopeptide would be elicited if compared to naive HT29 and MCF7 cell lines. Were comparisons to naive cell lines (untouched by DMSO) examined? It is necessary to show what effect DMSO is having on these cell lines' immunopeptidome to delineate the precise effect of birinapant.
4. Were any phenotypic differences observed in vivo experiments after administration of birinapant? Please provide this information if available, as it is important for readers to know if enhancing antigen presentation capabilities in vivo with birinapant changed tumor growth
5. Discussion section should include author's notes on differences in immunopeptidome observed with HT29 in vitro and in vivo
Author Response
- Please expand on current understanding of birinapant's effect on cancer cells both in vitro and in vivo
Answer: Thanks for this comment. We added this information in Introduction section. The contents are as follows:
Birinapant is a mimetic of second mitochondrial activator of caspases (SMAC). Previous studies on Birinapant focused on the activation of apoptotic pathways. In brief, Birinapant specifically binds to the baculoviral IAP repeat domain of Inhibitor of Apoptosis Proteins (IAP) through the bivalent AVPI tetrapeptide motif, then activates Caspases and promotes formation of apoptosome, at the end induces cell apoptosis. On the other hand, Birinapant also promotes the recruitment of NIK and activates the non-canonical nuclear factor-kappa B (Nf-κB) pathway. Birinapant has been shown to inhibit tumor cell proliferation by inducing apoptosis in a variety of tumor cell lines and xenograft models. For example, D L Zhu et al. found that Birinapant inhibits the invasion and proliferation of gastric cancer cells MGC-803 by promoting apoptosis (36). Jun Ding et al. found that Birinapant can promote apoptosis and inhibit invasion of liver cancer cell lines Huh7 and HepG2 (37). Najoua Lalaoui et al. found that Birinapant inhibited the proliferation of TNBC cells and PDX models through caspase-dependent apoptosis (38). Birinapant also functions in some combination drug strategies to enhance efficacy. Xuemei Xie et al. found that Birinapant targets IAP proteins in a variety of TNBC cell lines and CDX, inducing cell apoptosis, thereby enhancing the anti-tumor efficacy of gemcitabine in TNBC (39). David Cerna et al. found that Birinapant can also enhance the radiosensitivity of glioblastoma (40). In summary, for Birinapant, specific recognition and inactivation of IAP is a common feature, this is also a general understanding for many researchers on the drug function of SMAC mimetics.
- Why did the authors choose MCF7 cell line since we know TNBC is not a highly immunogenic cell line
Answer: Thanks for this comment. We chose MCF7 mainly because it was a common breast cancer cell, and breast cancer is the most common type of cancer in the world.
By the way, we check that MCF7 seemed belong to ER+, instead of TNBC.
- While DMSO is used as a vehicle control (necessary since it is the vehicle for birinapant) more accurate comparison of birinapant's effect on in vitro immunopeptide would be elicited if compared to naive HT29 and MCF7 cell lines. Were comparisons to naive cell lines (untouched by DMSO) examined? It is necessary to show what effect DMSO is having on these cell lines' immunopeptidome to delineate the precise effect of birinapant.
Answer: Thanks for this comment. I have to say sorry for we did not compare the immunopeptidomes between naïve cells and those treated with DMSO. We believe that BIR exists in a form dissolved in DMSO, so if we compare naïve cells with BIR treatment cells, in principle, the cellular environment contains two components, DMSO and BIR, so we cannot distinguish them. Therefore, in all experiments, we used simultaneously DMSO-treated samples to compare with BIR-treated samples. However, to evaluate the influence of DMSO, we used other methods (such as WB and QPCR) to detect changes in antigen presentation-related proteins in cells before and after DMSO or BIR treatment, which shows that compared to BIR, DMSO has a negligible effect on the antigen presentation of cells. By the way, what about your opinion on whether this figure added in supplementary or not?
The figure could be seen in pdf.
- Were any phenotypic differences observed in vivo experiments after administration of birinapant? Please provide this information if available, as it is important for readers to know if enhancing antigen presentation capabilities in vivo with birinapant changed tumor growth
Answer: Thanks for this comment. We observed slight changes in weight of mice and a significant reduction in tumor mass. This data is now displayed in F6. However, we need to point out that the reduction in tumor mass here is largely due to Birinapant-induced tumor cell apoptosis rather than host immune killing, because the mice we used are BALB/c nude, and their immune system is deficient.
- Discussion section should include author's notes on differences in immunopeptidome observed with HT29 in vitro and in vivo
Answer: Thanks for this comment. We added this content in Discussion section. The content is as follows:
We compared the immunopeptidomes of HT29 in vivo and in vitro. We found that the overlap was small. We believed this may be mainly caused by differences in the culture environment. One possible reason is that tumor cells face different survival pressures in vivo and in vitro, which may affect the immunopeptidome of the cells. It should be pointed out that all our immunopeptidome results come from three biological replicates, and the repeatability within the group seemed good, so it is unlikely that the low overlap is caused by erroneous identification. We found in one study that the immunopeptidomes of cells and xenograft models were highly reproducible (67). Therefore, it is also possible that the characteristics of the cells themselves cause this phenomenon. In addition, it may be related with the experiment method of the immunopeptidomics. We found that xx et al. used 1*109 cells and 1g tumor tissue for immunoprecipitation, which greatly exceeded amount of ours, but the number of immunopeptides they identified seemed smaller, especially in MDA-MB-436 and Colo205 cells, which may mean that a large number of peptides with low presentation abundance are lost. Overall, this issue still needs to be explored in more cell models to draw more credible conclusions.

Round 2
Reviewer 3 Report
Comments and Suggestions for Authors
1. Looks good
2. I apologize MCF7 is in fact ER+. It would be good to add a rationale to add why in vivo experiments were continued with HT29.
3. Yes, please mention in limitations that effect of DMSO on the immunopeptidome was not determined. Please add that figure in supplementary.
4 &5. Looks good now
Author Response
It would be good to add a rationale to add why in vivo experiments were continued with HT29.
Answer: Thanks for this comment. We add a sentence in the in vivo experiments part to explain this. The contents are as follows.
For obtain sufficient tumor samples for immunopeptidomes, we chose HT29 for model construction (MCF7 is more difficult to grow subcutaneously).
Yes, please mention in limitations that effect of DMSO on the immunopeptidome was not determined. Please add that figure in supplementary.
Answer: Thanks for this comment. We mention this in limitations:
Furthermore, although we demonstrated that compared with Birinapant, the effect on HLA of DMSO was negligible by western blot and FCM (Figure S11), the effect on immunopeptidomes of DMSO actually unclear and remains to be elucidated.
